# Atmospheric cloud-radiative heating in CMIP6 and observations, and its response to surface warming

Aiko Voigt[1], Stefanie North[1], Blaž Gasparini[1], and Seung-Hee Ham[2]

[1]Department of Meteorology and Geophysics, University of Vienna, Vienna, Austria
[2]Analytical Mechanics Associates (AMA), Hampton, VA, USA

**Correspondence:** Aiko Voigt (aiko.voigt@univie.ac.at)

**Abstract.** Cloud-radiation-interactions are key to Earth's climate and its susceptibility to change. While their impact on Earth's energy budget has been studied in great detail, their effect on atmospheric temperatures has received little attention, despite its importance for the planetary circulation of the atmosphere and hence for regional climate and weather. Here, we present the first systematic assessment of vertically-resolved cloud-radiative heating within the atmosphere in 20 CMIP6 models, including a comparison to satellite-based estimates. Our analysis highlights model differences in cloud-radiative heating in both the lower and upper troposphere, as well as uncertainties related to cloud ice processes. It also illustrates limitations of our ability to observe cloud-radiative heating. Not surprisingly, the response of cloud-radiative heating to surface warming is also uncertain across models. Yet, in the upper troposphere the response is very well predicted by an upward shift of the present-day heating, which we show results from the fact that cloud-radiative heating in the upper troposphere is a function of air temperature and thus decoupled from surface temperature. Our results have three important implications for upper-tropospheric cloud-radiative heating: they establish a new null hypothesis for its response to warming, offer a physics-based prediction of its response to warming based on present-day observations, and emphasize the need for improving its representation in simulations of the present-day climate, possibly by combining the benefits of upcoming km-scale models and satellite observations.

## 1 Introduction

The interactions of tiny cloud particles with even tinier photons are key to climate and its susceptibility to change. It is well understood that clouds regulate Earth's energy balance by scattering photons back to space in the shortwave domain of the electromagnetic spectrum, and by intercepting and re-emitting photons in the longwave domain (Ramanathan et al., 1989; Loeb et al., 2018). The interaction between clouds and radiation is often characterized by so-called cloud-radiative effects at the top of the atmosphere, which quantify the cloud impact by the difference between the energy balance of Earth and a hypothetical clear-sky climate in which clouds are assumed transparent to radiation. This widely-used top-of-atmosphere view forms the basis for much of our understanding of cloud-radiative feedbacks and their contribution to climate sensitivity (Sherwood et al., 2020). In this study, however, we take a different view and focus instead on cloud-radiation-interactions within the atmosphere, i.e., the radiative effect of clouds on atmospheric heating and cooling.

To quantify cloud-radiation-interactions within the atmosphere, we study the atmospheric cloud-radiative heating, a quantity sometimes also referred to as atmospheric cloud-radiative effects. Cloud-radiative heating is defined as

$$CRH(\lambda, \varphi, p, t) = \frac{\partial T}{\partial t}\bigg|_{\text{radiation}}^{\text{all-sky}} - \frac{\partial T}{\partial t}\bigg|_{\text{radiation}}^{\text{clear-sky}}. \tag{1}$$

The coordinates $(\lambda, \varphi, p, t)$ emphasize that $CRH$ is a time-varying function of pressure, or altitude, and geographical location. The first term of the right hand side of the equation is the radiative heating of the all-sky atmosphere that includes cloud-radiation-interactions. The second term is the radiative heating of a hypothetical clear-sky atmosphere that is identical to the all-sky atmosphere apart from the fact that clouds are not interacting with radiation. The philosophy behind cloud-radiative heating is the same as that behind top-of-atmosphere cloud-radiative effects, yet cloud-radiative heating is the more challenging quantity: very different vertical profiles of cloud-radiative properties and heating can lead to the same top-of-atmosphere cloud-radiative effects, and diagnosing cloud-radiative heating requires knowledge of the vertical distribution of clouds. It seems likely that this diagnostic challenge is one reason for why the vast majority of studies have focused on top-of-atmosphere cloud-radiative effects, while studies of cloud-radiative heating have remained relatively rare.

There is no shortage of reasons to understand cloud-radiative heating, however. Cloud-radiative heating influences the clouds that underlie its very existence, e.g., it prolongs the lifetime of tropical anvil clouds (Wall et al., 2020; Gasparini et al., 2022) and fuels the convective mixing that feeds subtropical marine low-level clouds (Stevens, 2005; Wood, 2012). Radiative cooling from low-level cloud tops drives tropical shallow overturning circulations (Naumann et al., 2019) and modulates the intensity of extratropical cyclones (Grise et al., 2019; Voigt et al., 2023). Radiative warming from tropical upper-tropospheric clouds narrows the intertropical convergence zone and decreases tropical-mean rainfall (Albern et al., 2018; Harrop and Hartmann, 2016). Cloud-radiative heating has also been shown to alter the internal variability of the climate system to varying degrees, including the Madden-Julian-Oscillation (Benedict et al., 2020), the El-Nino Southern Oscillation (Raedel et al., 2016) and the North Atlantic Oscillation (Li et al., 2014; Papavasileiou et al., 2020), and to be essential to the response of the planetary-scale circulation of the atmosphere to global warming (Voigt et al., 2019; Ceppi and Shepherd, 2017; Albern et al., 2019). Cloud-radiation interactions can further change the reservoir of available potential energy that is tapped by atmospheric motions (Stuhlmann and Smith, 1988; Romanski and Rossow, 2013; Kato et al., 2019). Given the range of scales on which cloud-radiative heating affects the atmospheric circulation, this list is clearly non-exhaustive. For a comprehensive review of the cloud-radiative heating impact on climate time scales we refer to Voigt et al. (2021); a discussion of the impact of cloud-radiative heating on extratropical weather systems can be found in Keshtgar et al. (2023).

A major motivation of our study is to provide a comprehensive assessment of atmospheric cloud radiative heating across global climate models. To this end, we compute and analyse cloud-radiative heating from model simulations of phase 6 of the Climate Model Intercomparison Project (Eyring et al., 2016). We are able to compute vertically-resolved cloud-radiative heating for 20 models. Previous model assessments were limited to a handful of models at most. Specifically, five models were compared in Cesana et al. (2019) and three models in Voigt et al. (2019); other studies only used one model or were restricted to specific regions (Johansson et al., 2021; Li et al., 2015). While Luo et al. (2023) used the CMIP6 model ensemble, they studied only the vertically-integrated heating derived from the difference in top-of-atmosphere and surface cloud-radiative fluxes.

We further compare models to satellite-based estimates of cloud-radiative heating from CERES-CALIPSO-CloudSat-MODIS (Kato et al., 2011; Ham et al., 2022) and CloudSat/CALIPSO (L'Ecuyer et al., 2008; Henderson et al., 2013).

Although previous assessments of cloud-radiative heating in models remained largely anecdotal, these studies collectively
provided evidence that cloud-radiative heating varies substantially between models and that poorly-constrained parameterizations such as cloud microphysics and aerosol-cloud-interactions strongly affect cloud-radiative heating. For example, using the ICON model with km-scale resolution over the Asian monsoon region, Sullivan and Voigt (2021) found that cloud-radiative heating in the tropical upper-troposphere changes by more than a factor of four when the model's 1-moment cloud microphysics scheme is replaced by a 2-moment scheme and microphysical information of the cloud particle effective radius is considered
in the radiation scheme. Similarly, using the ICON model over the North Atlantic region, Sullivan et al. (2023) reported strong sensitivity of cloud-radiative heating to cloud microphysics and the treatment of convection, and highlighted the influence of partitioning frozen hydrometeors into cloud ice, which is suspended in the atmosphere and taken into account in radiation calculations, and precipitating snow and ice, whose effects on radiation are typically neglected in models (Waliser et al., 2011; Li et al., 2022). Differences in cloud-radiative heating are also large between atmospheric reanalyses (Zhang et al., 2017; Wright
et al., 2020; Fujiwara et al., 2022), indicating that much of the challenge in correctly representing cloud-radiation-interactions in models does not stem from the large-scale circulation but small-scale cloud and radiative processes.

Studies with global climate models and atmospheric reanalyses have in particular highlighted shortcomings in upper-tropospheric cloud-radiative heating due to ice clouds (Voigt et al., 2019, 2021; Cesana et al., 2019; Johansson et al., 2021; Wright et al., 2020; Fujiwara et al., 2022). Ice clouds and their radiative heating are a key challenge for current models, both
at coarse resolutions of 100 km and fine km-scale resolutions (Gasparini et al., 2023). The challenging nature of ice clouds is unsurprising given the intricacies of their microphysical and radiative properties (Krämer et al., 2016; Zhang et al., 1999), yet understanding how they respond to warming is believed to be crucial to reduce uncertainty in climate sensitivity (Sherwood et al., 2020; Sokol et al., 2024) and to anticipate the response of the large-scale atmospheric circulation to warming (Voigt et al., 2019; Albern et al., 2019; Li et al., 2019).

While ice clouds are inherently complex, one important aspect of their response to warming is simple: ice clouds, or upper-tropospheric clouds more generally, tend to remain at roughly the same temperature as the climate warms. This behavior is an expression of the "fixed-anvil temperature hypothesis (FAT)" (Hartmann and Larson, 2002) that results from a well-understood thermodynamic control of convective outflow due to the rapid drop of clear-sky radiative cooling and water vapor near the tropopause. While originally developed for the tropics, FAT equally applies in the midlatitudes (Thompson et al.,
2017). FAT establishes an important expectation regarding the response of upper-tropospheric clouds to warming: because their temperature is constrained, ice clouds, and with them their radiative heating, will shift upward as the surface warms. Modeling and observational analyses unequivocally support an upward shift of upper-tropospheric clouds in response to surface warming (Kuang and Hartmann, 2007; Norris et al., 2016; Po-Chedley et al., 2019; Richardson et al., 2022; Zelinka and Hartmann, 2011; Zelinka et al., 2016). The upward shift suggests that the response of upper-tropospheric cloud-radiative heating to warming can
be "predicted" by an upward shift of the present-day cloud radiative heating. This idea is supported by the findings of Singh and O'Gorman (2012) and Voigt et al. (2019); here we show it also holds across the CMIP6 model ensemble. This establishes

a strong constraint on the global warming response of ice clouds, and hence their radiative impact on atmospheric circulation and regional climate change.

Throughout the paper, our focus is on the representation of cloud-radiative heating within the atmosphere in global climate models for two reasons. First, by contributing to the diabatic heating of the atmosphere, cloud-radiative heating is relevant to the atmospheric circulation. And second, a systematic study of cloud-radiative heating in models is lacking, in contrast to cloud mass and fraction (Li et al., 2012; Lauer et al., 2023). Because cloud-radiative heating is defined as the difference between all-sky and clear-sky radiative heating, it depends not only on the cloud field itself but also the clear-sky background state of the atmosphere, an effect known as "cloud-masking" (Soden et al., 2004; Huang and Huang, 2024). Differences in cloud-radiative heating between models, between models and observations, or between different climate states may thus be influenced by non-cloud fields such as temperature and water vapor. These clear-sky effects could be quantified by explicit radiative transfer calculations or radiative kernel methods, but given the paucity of studies on atmospheric cloud-radiative heating we leave such refinements to future work.

Our manuscript is organized as follows. Sect. 2 describes the calculation of cloud-radiative heating in CMIP6 models and observational estimates of cloud-radiative heating in today's climate. Sect. 3 studies cloud-radiative heating in simulations of present-day climate. Sects. 4 and 5 address how cloud-radiative heating responds to uniform and non-uniform surface warming. Sect. 6 asks to what extent the response can be predicted from the present-day climate. The paper concludes in Sect. 7, where we articulate three important consequences of our work.

## 2 CMIP6 simulations and satellite-based estimates of CRH

### 2.1 CMIP6 simulations

We use model output from the amip, amip-p4K and amip-future4K simulations of CMIP6 (Eyring et al., 2016; Webb et al., 2017). In the amip simulation, sea-surface temperatures, sea ice, well-mixed greenhouse gases and aerosols are prescribed to observed values from 1979 to 2014. This enables a clean comparison between models as well as between models and observations. In the amip-p4K simulation, sea-surface temperatures are uniformly increased by 4 K. In the amip-future4K simulation, sea-surface temperatures are also increased by 4 K in the global mean, but the increase varies spatially according to a pattern derived from coupled climate models. The amip-p4K and amip-future4K simulations allow us to study the response of cloud-radiative heating to surface warming and to assess to what extent the response depends on the pattern of surface warming.

We use amip simulations from 20 CMIP6 models for which we were able to retrieve the all-sky and clear-sky radiative fluxes or heating rates from the CMIP6 ESGF archive that are necessary to calculate cloud-radiative heating, as described in the following subsection. The models are listed in Tab. 1. Seven of the 20 models moreover provide the necessary output for the amip-p4K and amip-future4K simulations. We further retrieve atmospheric temperature, based on which we calculate the thermal tropopause as defined by the World Meteorological Organization (WMO) using the PyTropD python package of Adam et al. (2018). All data retrieval and analysis scripts are included in the accompanying data.

## 2.2 Calculation of cloud-radiative heating from CMIP6 model output

According to Eq. 1 cloud-radiative heating is given as the difference in all-sky and clear-sky radiative heating rates within the atmosphere. Since radiative heating rates are given by the radiative flux divergence divided by the mass of air, cloud-radiative heating can be calculated either directly from heating rates or indirectly from radiative fluxes:

$$CRH \quad = \quad \left.\frac{\partial T}{\partial t}\right|_{\text{radiation}}^{\text{all-sky}} - \left.\frac{\partial T}{\partial t}\right|_{\text{radiation}}^{\text{clear-sky}} \tag{2}$$

$$= \quad -\frac{g}{c_p} \cdot \frac{\partial}{\partial p}\left(F^{\text{all-sky}} - F^{\text{clear-sky}}\right) \tag{3}$$

$$= \quad \frac{1}{\rho c_p} \cdot \frac{\partial}{\partial z}\left(F^{\text{all-sky}} - F^{\text{clear-sky}}\right). \tag{4}$$

Here, $\rho$ is the air density, $c_p$ is the heat capacity of air at constant pressure, $g$ is gravitational acceleration, and $F$ denotes the radiative fluxes in all-sky and clear-sky conditions, respectively (fluxes are defined as positive downward). We follow three approaches to calculate cloud-radiative heating, depending on the available output:

1. If the full set of all-sky and clear-sky radiative fluxes is available from the CFmon table, we first calculate individual heating rates from the radiative flux divergence following Eqs. 3 and 4, respectively, and then calculate cloud-radiative heating from the heating rates using Eq. 2. Eq. 3 is used for models with pressure-based vertical coordinates, Eq. 4 for models with height-based vertical coordinates.

2. If all-sky and clear-sky radiative heating rates are available from the CFmon, AERmon or Emon tables, we calculate cloud-radiative heating directly from the difference in heating rates using Eq. 2.

3. If the zonally-averaged radiative heating rates are provided as part of the EmonZ table, we use these to calculate cloud-radiative heating from Eq. 2.

For approaches 1 and 2, cloud-radiative heating is calculated on model levels and subsequently interpolated vertically to a set of 100 common pressure levels from 1000 to $0\,\text{hPa}$ with a level spacing of $10\,\text{hPa}$. For approach 1 it is essential to derive heating rates from radiative fluxes on model levels; computing heating rates from fluxes interpolated to pressure levels can result in substantial errors. This is illustrated in the additional figure A1. Cloud-radiative heating from approach 3 is interpolated from the EmonZ pressure levels to the common pressure levels. All calculations use monthly mean model output. The heat capacity of dry air, $c_p = 1005\,\text{Jkg}^{-1}\text{K}^{-1}$, is used; the impact of humidity on $c_p$ is well below a few percent (Rogers and Yau, 1989) and can be neglected for the purpose of our study. The additional figures A2, A3 and A4 illustrate cloud-radiative heating calculated from the three approaches, showing that approaches 1 and 2 give indistinguishable results. For the analysis, we preferably use cloud-radiative heating calculated by approach 1 because it is applicable to the majority of the models. If approach 1 is not possible, we use approach 2, if possible, or approach 3.

**Table 1.** CMIP6 models for which cloud-radiative heating is calculated. The approach used to calculate cloud-radiative heating depends on the data availability and is indicated by the checkmarks. See the manuscript text for an explanation of the three approaches.

| Simulation | Model | Radiative fluxes (approach 1) | Temperature tendencies (approach 2) | Zonal-mean temperature tendencies (approach 3) |
|---|---|---|---|---|
| amip | BCC-CSM2-MR | ✓ | | |
| | CESM2 | | | ✓ |
| | CESM2-FV2 | | | ✓ |
| | CESM2-WACCM | | | ✓ |
| | CESM2-WACCM-FV2 | | | ✓ |
| | CNRM-CM6-1 | ✓ | ✓ | |
| | CNRM-ESM2-1 | ✓ | ✓ | |
| | EC-Earth3 | | | ✓ |
| | GFDL-AM4 | | ✓ | |
| | GFDL-CM4 | | ✓ | |
| | GFDL-ESM4 | | | ✓ |
| | HadGEM3-GC31-LL | ✓ | ✓ | |
| | HadGEM3-GC31-MM | ✓ | | |
| | INM-CM4-8 | ✓ | | |
| | INM-CM5-0 | ✓ | | |
| | IPSL-CM6A-LR | ✓ | | |
| | MIROC-ES2L | ✓ | | |
| | MIROC6 | ✓ | | |
| | MRI-ESM2-0 | ✓ | | |
| | UKESM1-0-LL | ✓ | ✓ | |
| amip-p4K | BCC-CSM2-MR | ✓ | | |
| | CNRM-CM6-1 | ✓ | | |
| | GFDL-CM4 | | ✓ | |
| | HadGEM3-GC31-LL | ✓ | ✓ | |
| | IPSL-CM6A-LR | ✓ | | |
| | MIROC6 | ✓ | | |
| | MRI-ESM2-0 | ✓ | | |
| amip-future4K | BCC-CSM2-MR | ✓ | | |
| | CNRM-CM6-1 | ✓ | | |
| | GFDL-CM4 | | | ✓ |
| | HadGEM3-GC31-LL | ✓ | ✓ | |
| | IPSL-CM6A-LR | ✓ | | |
| | MIROC6 | ✓ | | |
| | MRI-ESM2-0 | ✓ | | |

Two notes are in order for the models HadGEM3-GC31-LL, HadGEM3-GC31-MM and UKESM1-0-LL, which differ from the rest of the models by using a vertical grid based on height. The first note concerns the calculation of cloud-radiative heating from the flux divergence using Eq. 4. The calculation requires the factor $\rho \cdot c_p$ that depends on time, latitude, longitude and model level. We obtain this factor from the ratio of the all-sky radiative heating rate and the all-sky radiative flux as $\rho \cdot c_p = \partial_z F^{\text{all-sky}} / \partial_t T|^{\text{all-sky}}_{\text{radiation}}$. The factor is then used to convert radiative fluxes to heating rates for all components of radiation, i.e., shortwave and longwave as well as clear-sky and all-sky fluxes. The second note concerns the vertical interpolation. The heating rates for the three model are interpolated to a common height grid extending from 0 to 20 km with a level spacing of 200 m. Where needed, e.g., for plots across all models, we convert from altitude levels to pressure levels using the geopotential height variable zg from the Amon table.

## 2.3 Satellite-based estimates of cloud-radiative heating

We compare the amip simulations to two estimates of cloud-radiative heating based on active satellite measurements. Kato et al. (2019) demonstrated the need for active measurements of clouds to derive radiative heating rates within the atmosphere. The first estimate is from the RelD1 product of CERES-CALIPSO-CloudSat-MODIS (CCCM; Kato et al., 2021), the second estimate from the 2B-FLXHR-LIDAR product of CloudSat/CALIPSO (L'Ecuyer et al., 2008; Henderson et al., 2013). Both products combine the satellite measurements with radiative transfer calculations to derive all-sky and clear-sky radiative heating rates, from which cloud-radiative heating can be calculated. For the CCCM estimate, we follow the procedure described in Ham et al. (2017) and derive heating rates on height levels from daytime radiative fluxes following Eq. 4, with shortwave heating rates scaled by monthly gridded solar incoming flux from the CERES SYN product. For the 2B-FLXHR-LIDAR estimate, we use the release R05 and derive cloud-radiative heating following the approach described in Papavasileiou et al. (2020): heating rates are derived for each CloudSat/CALIPSO granule using Eq. 3 and ancillary information from the ECMWF-AUX, and then binned into 2.5 deg lon x 2.5 deg lat averages for time periods of 5 days. Similar to the CCCM estimate, shortwave heating rates are scaled by daily-mean radiative insolation obtained from the CLIMLAB python package of Rose (2018). We average the pentad values in time to obtain the climatological cloud-radiative heating. We also derive cloud-radiative heating for the earlier R04 release of 2B-FLXHR-LIDAR analyzed by Papavasileiou et al. (2020), but if not noted otherwise we use the R05 release. The comparison between the estimate from CCCM and the two releases of 2B-FLXHR-LIDAR allows us to illustrate current uncertainties in observing cloud-radiative heating.

For both CCCM and 2B-FLXHR-LIDAR, we analyze the time period from 2007-2010 and convert from altitude to pressure levels by means of the MetPy implementation (May et al., 2022) of the U. S. standard atmosphere (National Oceanic and Atmospheric Administration, National Aeronautics and Space Administration, and U. S. Air Force, 1976).

## 2.4 Cloud fraction

To allow the reader to connect the spatial pattern of cloud-radiative heating to cloud fraction, we include the latter in some of our figures. For the models, cloud fraction is taken from the variable cl of the Amon table and interpolated from model levels to pressure levels in the same manner as radiative heating rates. For observations, we use the CloudSat/CALIPSO based

community product of Bertrand et al. (2024) for years 2007-2010. It should be noted, however, that cloud-radiative heating also depends on cloud liquid and ice content and the radiative treatment of clouds (e.g., Keshtgar et al., 2024) and may be affected by cloud masking (cf. Sect. 1). Differences in cloud-radiative heating between models and between models and observations are thus not explained by cloud fraction alone.

## 3 Cloud-radiative heating in the present-day climate

We begin with the zonal-mean time-mean cloud-radiative heating in the amip simulations shown in Fig. 1. The models agree on the overall pattern of cloud-radiative heating: clouds radiatively heat the upper troposphere at low latitudes and cool the lower troposphere, consistent with the meridional distribution of high-level and low-level clouds shown in Fig. 2. The two figures illustrate the well-known pattern of warming in the lower part of the cloud and cooling in its upper part that results from the absorption and emission of longwave radiation (cf. Fig. 3 of Slingo and Slingo, 1988). This dipole pattern is for example evident for low-level clouds in the subtropics and extratropics and high-level clouds in the extratropical storm track regions. For tropical high-level clouds, the upper part of the cloud is warming instead of cooling because shortwave heating is larger and the low optical thickness that is typical for many high-level tropical clouds reduces their emission of longwave radiation.

Beyond this basic agreement, however, the models show substantial differences in cloud-radiative heating at all latitudes. In the tropics, the maximum radiative heating in the upper troposphere ranges from $0.3\,\mathrm{K\,day^{-1}}$ in the family of CESM2 models to $0.9\,\mathrm{K\,day^{-1}}$ in the family of GFDL models, and the vertical distribution of cloud-radiative heating ranges from a clear maximum in the upper troposphere, e.g., in the GFDL and MIROC models, to a rather uniform heating within the free troposphere, e.g., in the CNRM and INM models. In the extratropics, between 30 and 60 deg latitude, the maximum cloud-radiative cooling in the lower troposphere varies by a factor of four between $-0.4\,\mathrm{K\,day^{-1}}$ in EC-Earth3 to $-1.6\,\mathrm{K\,day^{-1}}$ in IPSL-CM6A-LR, and while some models show a dipole of cloud-radiative cooling near the tropopause and heating below (e.g., BCC-CSM2-MR and the two MIROC models), cloud-radiative heating is negative or close to zero throughout the extratropical free troposphere in other models (e.g., IPSL-CM6A-LR as well as the CESM and INM models).

To quantify the model differences in cloud-radiative heating, the top row of Fig. 3 shows the model median and the standard deviation across models. Consistent with the findings above and with findings from the five GASS-YOTC models (Jiang et al., 2015; Klingaman et al., 2015) compared in Cesana et al. (2019), the figure highlights two regions. First, model differences are large in the lower troposphere, in particular in the Southern Hemisphere extratropics, where mixed-phase clouds dominate and the standard deviation across models is almost as large as the model median. Second, model differences are pronounced in the upper troposphere around the -38 deg C isotherm, where ice clouds prevail as a consequence of convective motions and large-scale ascent. In comparison, model differences are small in the mid troposphere.

Model differences in cloud-radiative heating are not simply a result of clear-sky differences but truly reflect differences in the radiative interactions of clouds and translate to model differences in all-sky radiative heating. This is illustrated in Fig. 4, which shows the variance in all-sky, clear-sky and cloud-radiative heating across all amip models. Mathematically, the variance

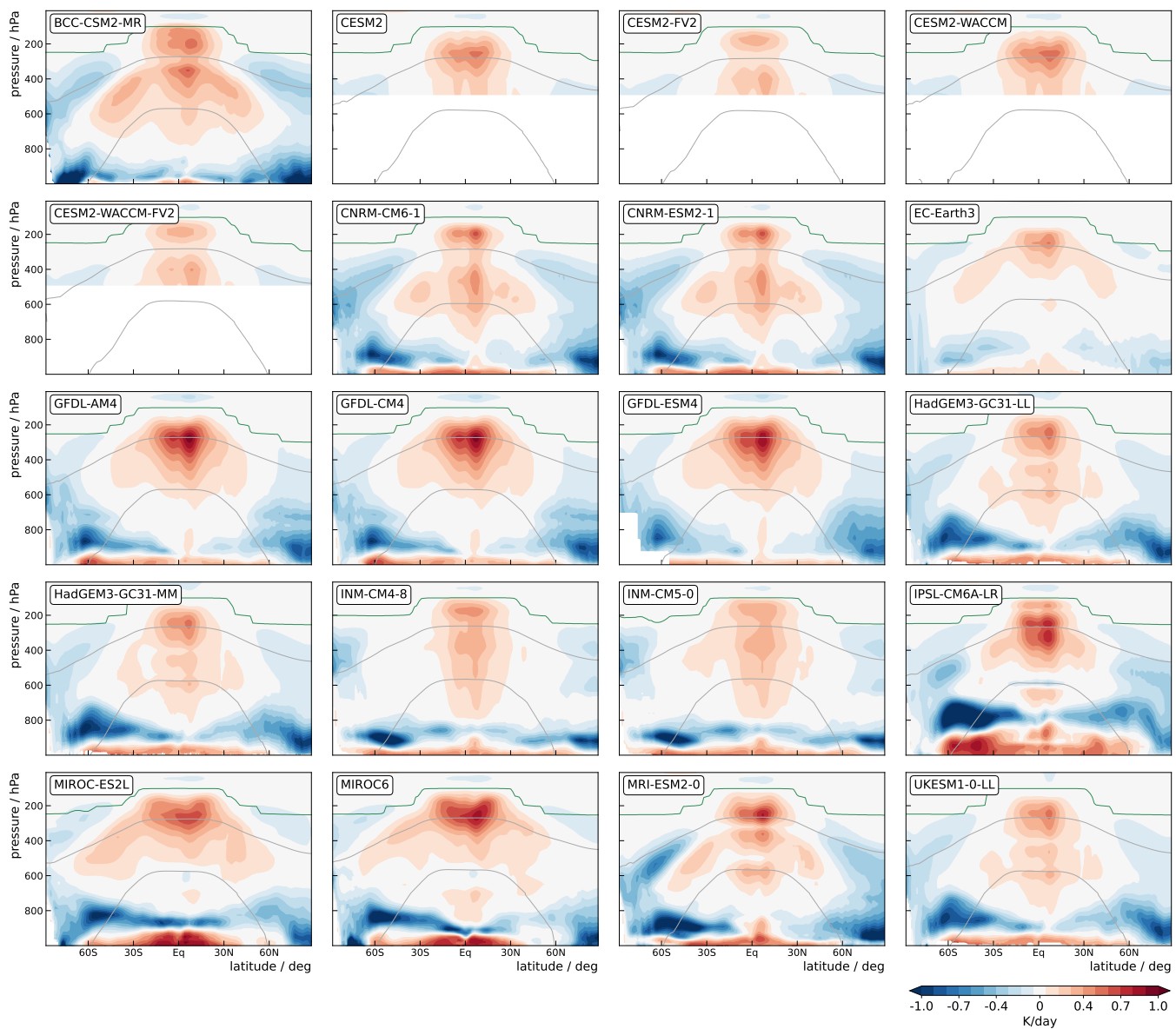

**Figure 1.** Zonal-mean time-mean cloud-radiative heating in amip simulations of 20 CMIP6 models. The gray lines mark the 0 and -38 deg C isotherms to loosely distinguish regions of liquid, mixed-phase and ice clouds. The green line marks the thermal tropopause. Cloud-radiative heating in the CESM2 models is not shown below 500 hPa because their EmonZ model output is affected by surface topography (cf. additional figure A2).

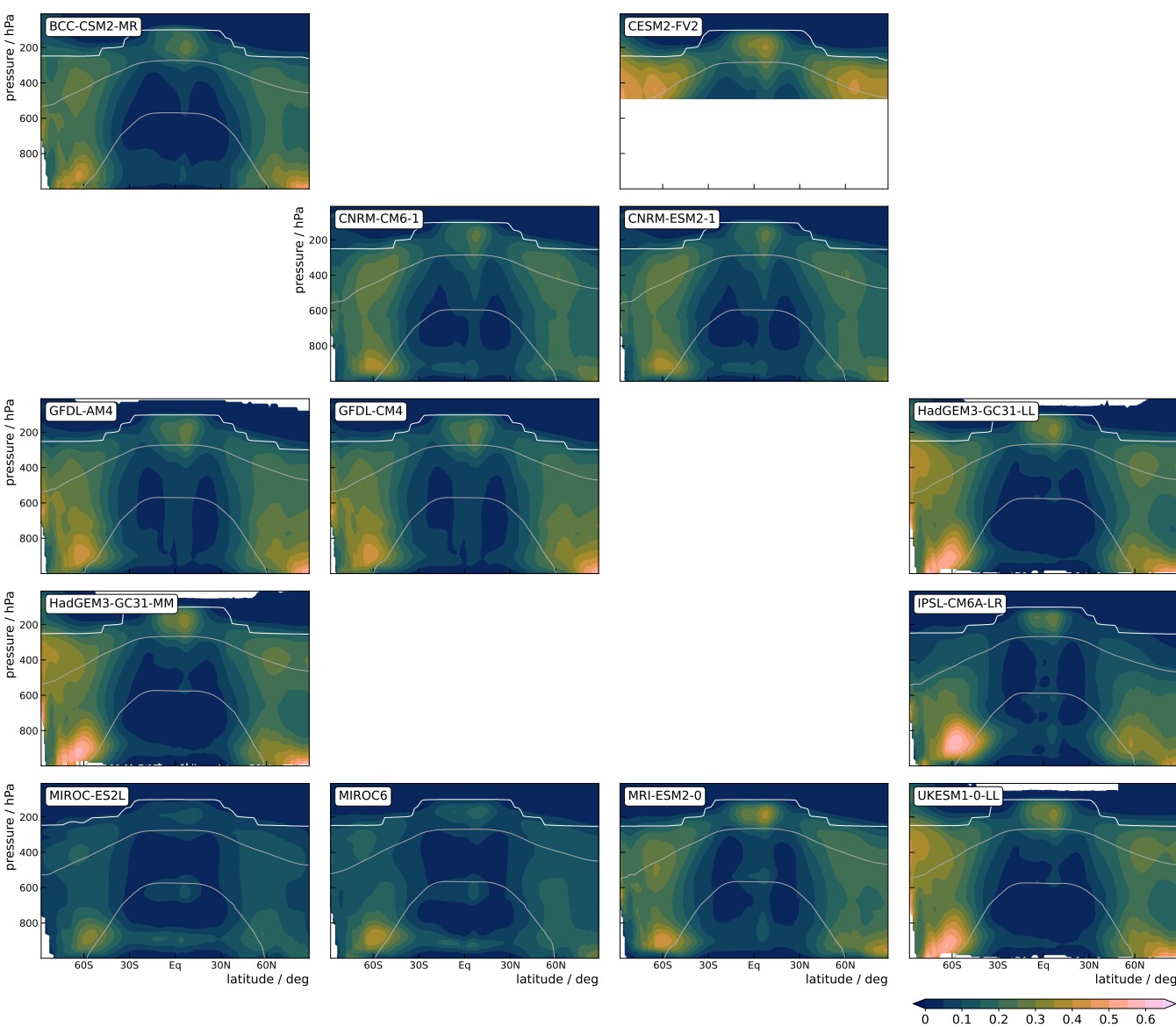

**Figure 2.** Zonal-mean time-mean cloud fraction in amip simulations. The gray lines mark the 0 and -38 deg C isotherms to loosely distinguish regions of liquid, mixed-phase and ice clouds. The white line marks the thermal tropopause. Note that for some models cloud fraction is not available from the ESGF archive; for CESM2-FV2 cloud fraction is restricted to above 500 hPa.

in all-sky heating, $\sigma^2_{\text{all-sky}}$, can be decomposed into the variances in cloud and clear-sky heating and their covariance,

$$\sigma^2_{\text{all-sky}} = \sigma^2_{\text{cloud}} + \sigma^2_{\text{clear-sky}} + 2 \cdot \text{cov}(\text{cloud}, \text{clear-sky}). \tag{5}$$

Model differences in all-sky radiative heating overall are dominated by cloud-radiative heating. Moreover, although differences in cloud-radiative heating are in some regions co-related to clear-sky differences (e.g., in the tropical upper troposphere), they do not appear to be driven primarily by the latter. Future work that takes into account cloud-masking effects would be helpful to quantify the sources of model differences in all-sky and cloud-radiative heating, for example by means of radiative kernels (Huang and Huang, 2023, 2024).

To put the model simulations into perspective, Fig. 3 also shows the satellite-based estimates of cloud-radiative heating in panels c-e, together with the observed cloud fraction in panel f. A somewhat sobering aspect is that the difference between satellite-based estimates of cloud-radiative heating from 2B-FLXHR-LIDAR and CCCM is about as large as the standard deviation across models. Ham et al. (2017) showed that assumptions regarding cloud detection thresholds, cloud merging and cloud optical properties substantially affect cloud-radiative heating rates and explain the differences between 2B-FLXHR-LIDAR and CCCM. For example, in the tropical upper troposphere, cloud-radiative heating is more positive in CCCM because the larger ice cloud extinction coefficient and particle size leads to stronger absorption in both the shortwave and longwave domains. The differences between R05 and R04 of 2B-FLXHR-LIDAR result from improvements in cloud microphysics, surface albedo and cloud detection (CloudSat Project, 2023). Overall, the differences in the satellite-based estimates illustrate the challenges in establishing an observational baseline for cloud-radiative heating.

Fig. 5 characterizes the cloud-radiative heating by means of vertical profiles over five domains that separate tropical ascending motion (15 deg S - 15 deg N) from subtropical descending motion (15 deg N/S - 35 deg N/S) and extratropical regions dominated by midlatitude cyclones (35 deg N/S - 70 deg N/S). The model simulations are shown in gray, the satellite-based estimates in blue. Again, the lower and upper troposphere emerge as regions with large differences between models as well as between the satellite-based estimates, while the differences are smaller in the mid-troposphere. In the Southern hemisphere extratropics and in the subtropics of both hemispheres, low-level cloud-radiative cooling in most of the models peaks at too low altitudes compared to the satellite-based estimates, consistent with Cesana et al. (2019). High-level cloud-radiative heating in the tropical upper troposphere is lower in the models compared to the CCCM estimate, in contrast to Cesana et al. (2019).

## 4   Response to uniform ocean surface warming

We now study the response of cloud-radiative heating to climate change. To this end, we analyze the amip-p4K simulations in which the ocean surface is warmed uniformly by 4 K. We will show that the response of cloud-radiative heating to surface warming differs markedly between models, and that the model differences in the upper troposphere are nearly entirely caused by model differences in the present-day climate. The latter leads to the possibility to predict the response of upper-tropospheric cloud-radiative heating to surface warming based on the present-day climate.

Fig. 6 shows the response of zonal-mean time-mean cloud radiative heating in the amip-p4K simulation relative to the amip simulations. Note that only seven models provide the output to calculate cloud-radiative heating in the amip-p4K simulations.

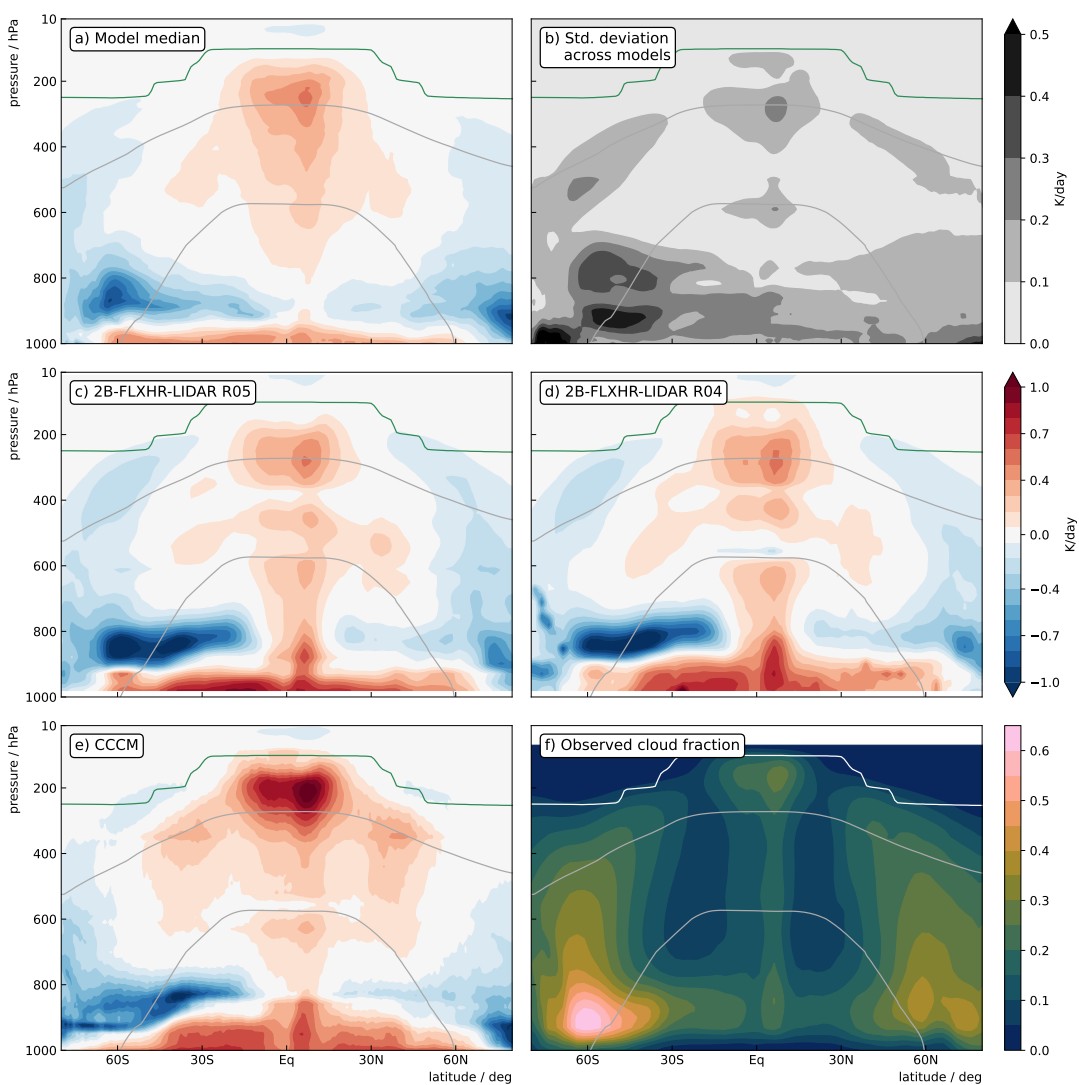

**Figure 3.** Agreement and disagreement in cloud-radiative heating in models and observations. The upper row show the model median (panel a) of zonal-mean time-mean cloud-radiative heating and the standard deviation (panel b) across the amip simulations of the 20 CMIP6 models. The middle and lower rows shows the satellite-based estimates of cloud-radiative heating from the 2B-FLXHR-LR R05 and R04 (panels c and d) and CCCM (panel e). Observed cloud fraction is shown in panel f. In all panels, the gray lines mark the model median of the 0 and -38 deg C isotherms and the green line marks the model median of the thermal tropopause (white line in panel f).

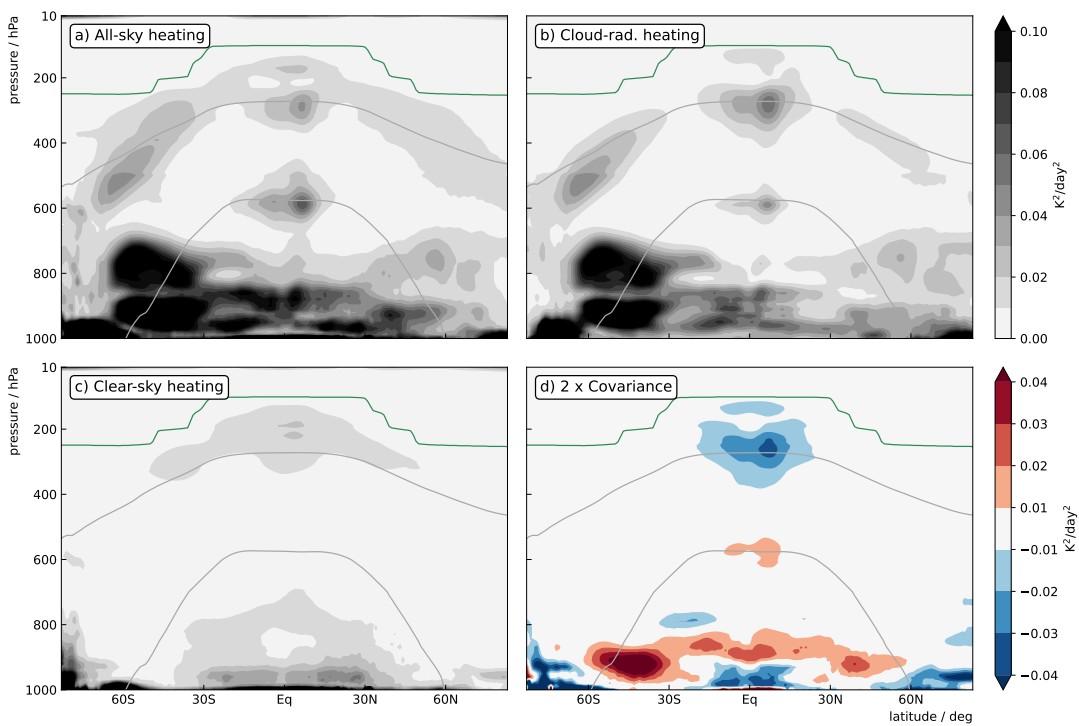

**Figure 4.** Variance across amip model simulations for zonal-mean time-mean all-sky, cloud and clear-sky radiative heating in panels a-c. Panel d shows the covariance between the cloud and clear-sky radiative heating across models, multiplied by a factor of two. By design, panel a is the sum of panels b-d.

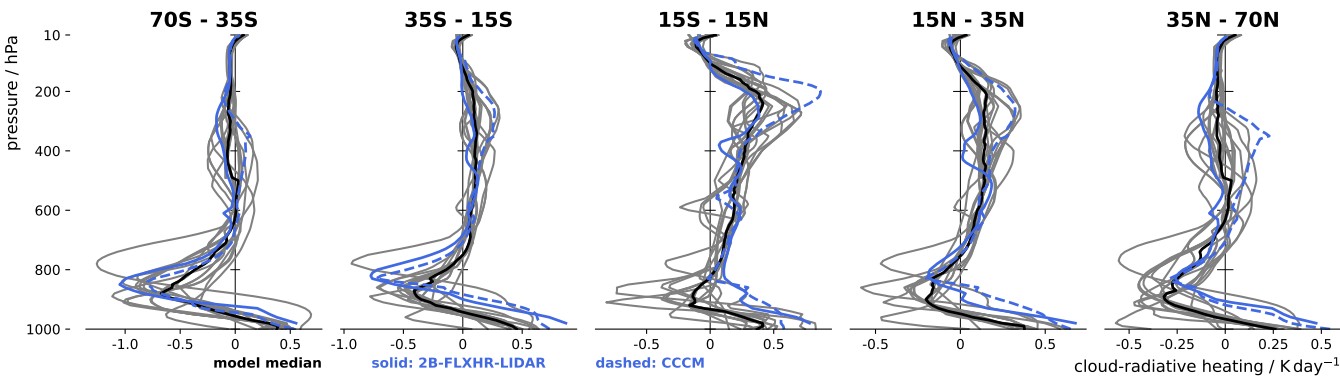

**Figure 5.** Time-mean cloud-radiative heating averaged over the domains of the extratropical storm track (70-35 deg lat), subtropical descent (35-15 deg lat) and tropical ascent (equatorward of 15 deg lat). The black line shows the model median. The blue lines show the cloud-radiative heating estimated by 2B-FLXHR-LIDAR R05 (solid) and CCCM (dashed).

The response shows a very complicated pattern, and it is difficult to identify aspects that robustly emerge in models. This is in particular the case in the lower troposphere, where the response of cloud-radiative heating is tied to model-dependent changes in low-level warm clouds in the tropics and subtropics and low-level mixed-phase clouds in the extratropics.

In the upper troposphere, however, cloud-radiative heating responds to surface warming in a manner that is more robust across models. Despite large differences between models, all models simulate an arc-like pattern of anomalous positive cloud-radiative heating that extends from the tropics into the extratropics and is roughly aligned with the -38 deg C isotherm. For example, the arc is very well developed in the BCC-CSM2-MR model. Models differ in terms of the extent of the arc towards the poles as well as its altitude and position relative to the -38 deg C isotherm, yet they agree that the upper-tropospheric

response of cloud-radiative heating includes the arc-like pattern. In fact, this pattern was reported previously by Voigt and Shaw (2016); Voigt et al. (2019) and Voigt et al. (2021), who attributed it to the upward extension of the tropopause and the associated upward shift of clouds.

       We now demonstrate that the upper-tropospheric response of cloud-radiative heating response, i.e., above 500 hPa, is very well predicted from an upward shift of cloud-radiative heating simulated in the models for the present-day climate. The predic-

tion is anchored in the fact that upper-tropospheric clouds are strongly tied to atmospheric temperature (see also further below) via the fixed-anvil temperature hypothesis (Hartmann and Larson, 2002; Thompson et al., 2017). The prediction follows Singh and O'Gorman (2012), who developed a framework to describe the upward extension of the tropopause with warming by rescaling the pressure coordinate. Eq. 13g of Singh and O'Gorman (2012) in particular predicts the upward shift of atmospheric radiative heating by shifting the pressure levels $p$ to new pressure levels $p' = p/\beta$. $\beta$ is larger than 1, which implies

$p' < p$ and thus an upward shift. With this, the response of cloud-radiative heating can be predicted as

$$dCRH(p,\varphi) = CRH(p,\varphi) - CRH(\beta p,\varphi). \tag{6}$$

$\beta$ depends on the magnitude of surface warming. To derive its value, we compute the warming in the amip-p4K simulations at 800 hPa averaged between 70 deg N/S, yielding a model mean of 4.8 K. Using a rounded value of 5 K and Eq. 15 of Singh and O'Gorman (2012), we obtain $\beta = 1.2$. We use the same value for all models to highlight that the knowledge of the surface warming is sufficient to predict the upward shift of cloud-radiative heating.

Fig. 7 shows the cloud-radiative heating response averaged over the five extratropical, subtropical and tropical domains of Fig. 5. The upper row includes all seven models and illustrates the large model differences in the response. The rows below show the individual models as well as the predicted response as blue lines. For almost all models and domains, the prediction captures the actual response extremely well. This is not only evident from the visual inspection of Fig. 7 but also from the correlation coefficients between the vertical profiles of the actual and the predicted responses. The correlation coefficients are

calculated as the Pearson product-moment correlation coefficients and are typically larger than 0.9. We note that while the prediction has been successfully applied previously in Voigt et al. (2019) for two of their three considered models, here we show that indeed it holds broadly across global climate models.

       A drawback of the domain averages on pressure levels is that they tend to obscure the dependence of clouds on atmospheric temperature that underlies the successful prediction in Fig. 7. Another drawback is that domain averages on pressure levels are

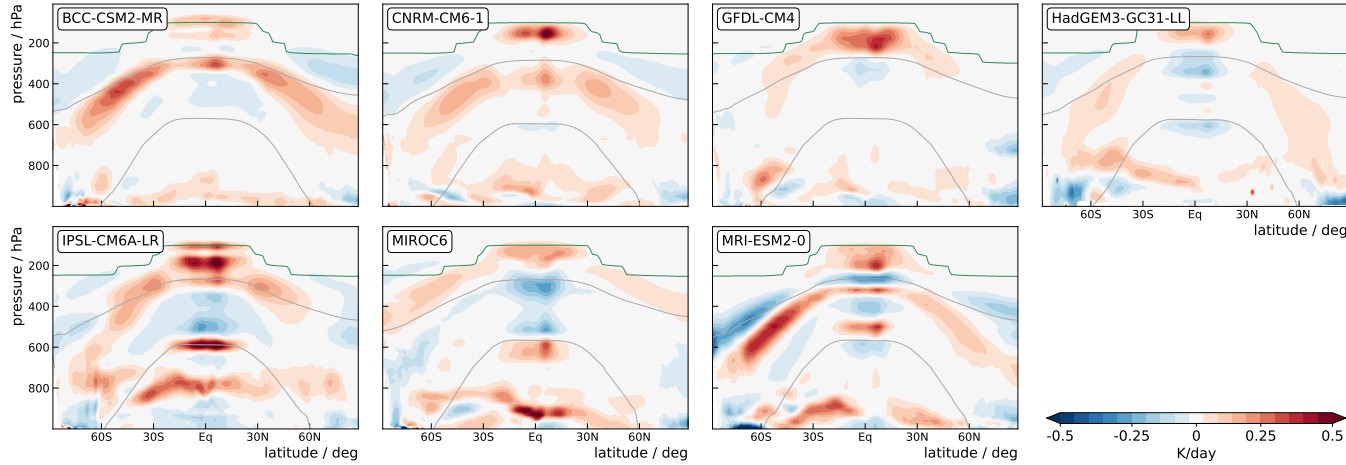

**Figure 6.** Response of zonal-mean time-mean cloud-radiative heating to a uniform 4 K warming of the ocean surface in amip-p4K simulations with 7 CMIP6 models. The gray lines mark the 0 and -38 deg C isotherms and the green lines the thermal tropopause for the amip simulation.

affected by a cancellation between positive and negative values of cloud-radiative heating in the extratropics, where meridional temperature gradients are strong and isotherms and isobars are not aligned with each other (see, e.g., Figs. 1 and 6). To address these drawbacks, we also calculate cloud-radiative heating as a function of atmospheric temperature, i.e., we use temperature instead of pressure as the vertical coordinate. When remapping from pressure levels to temperature levels, we use the zonal-mean time-mean temperature and only consider levels below the tropopause since the increase in temperature above the tropopause makes the mapping non-unique. For reference, the zonal-mean time-mean cloud-radiative heating in the amip simulations sampled as a function of temperature is shown in additional figure A5. Here, we focus on the response of cloud-radiative heating to surface warming in the amip-p4K simulations.

Fig. 8 shows that apart from the region of the boundary layer, the cloud-radiative heating is remarkably similar between the amip and amip-p4K simulations when expressed as a function of temperature. In 5 of the 7 models, the difference in cloud-radiative heating between the two climates is within $\pm 0.1$ K/day and hence close to zero. In IPSL-CM6A-LR and CRNM-CM6-1, cloud-radiative heating increases by roughly 30% in the tropical mid and upper troposphere, yet the increase is closely aligned with the cloud-radiative heating in the present-day climate. This suggests that it results largely from an increase in cloud ice content or cloud fraction. Although future work should address why cloud-radiative heating increases in these two models but not in the other models, our analysis clearly establishes a helpful null hypothesis: cloud-radiative heating does not change with warming apart from an upward shift to lower pressures so as to stay at the same air temperature.

Expressing cloud-radiative heating as a function of temperature thus shows that the upward shift is an excellent prediction of the response of upper-tropospheric cloud-radiative heating to surface warming because clouds and their radiative heating are invariant to climate change when viewed not in terms of pressure but air temperature. This view is in line with the work

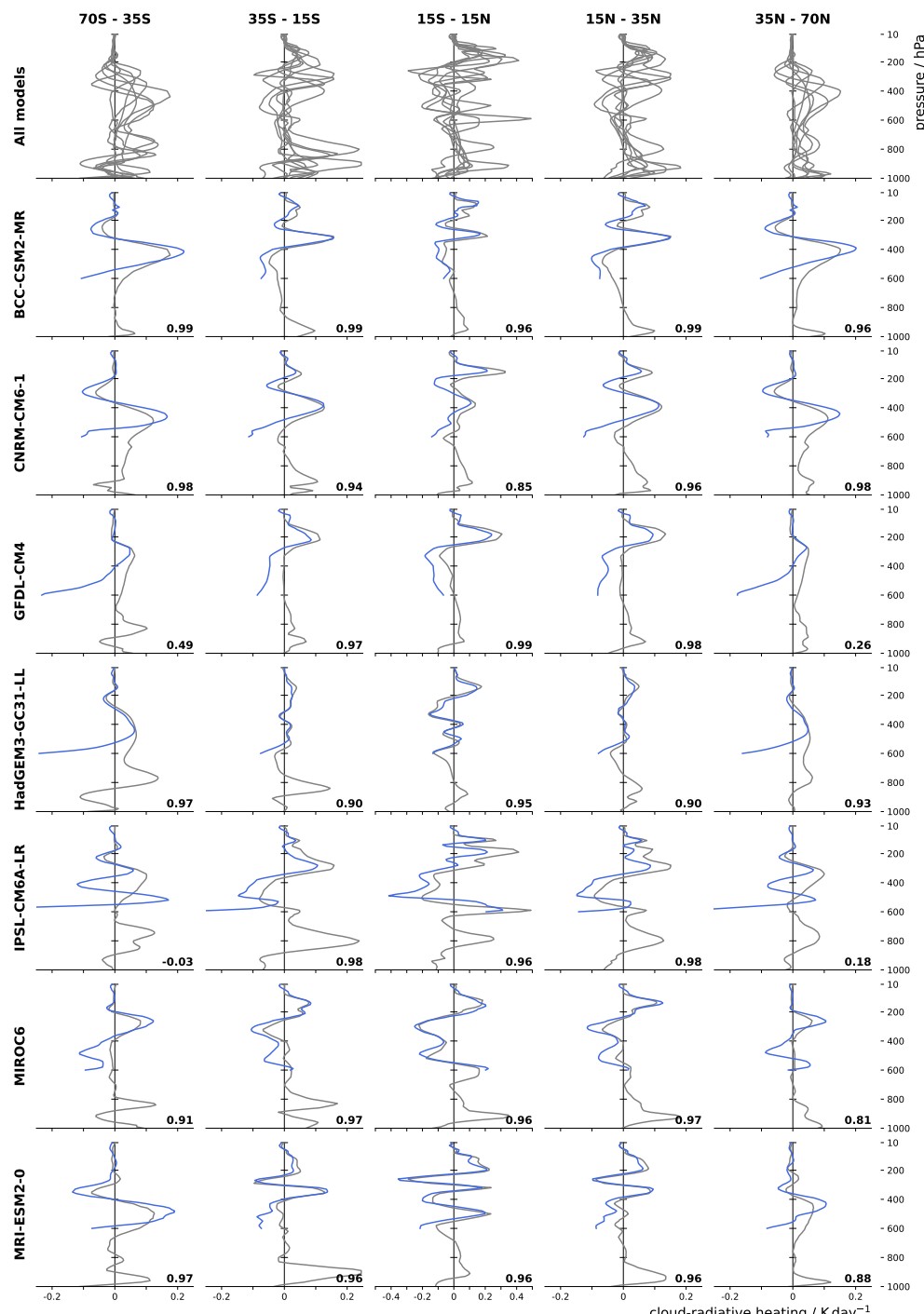

**Figure 7.** Response of cloud-radiative heating to surface warming averaged over the domains of the extratropical storm track (70-35 deg lat), subtropical descent (35-15 deg lat) and tropical ascent (equatorward of 15 deg lat). The blue line shows the response predicted by the upward shift of cloud-radiative heating in the present-day climate. The numbers give the correlation coefficient between the actual and the predicted response between 100 and 500 hPa.

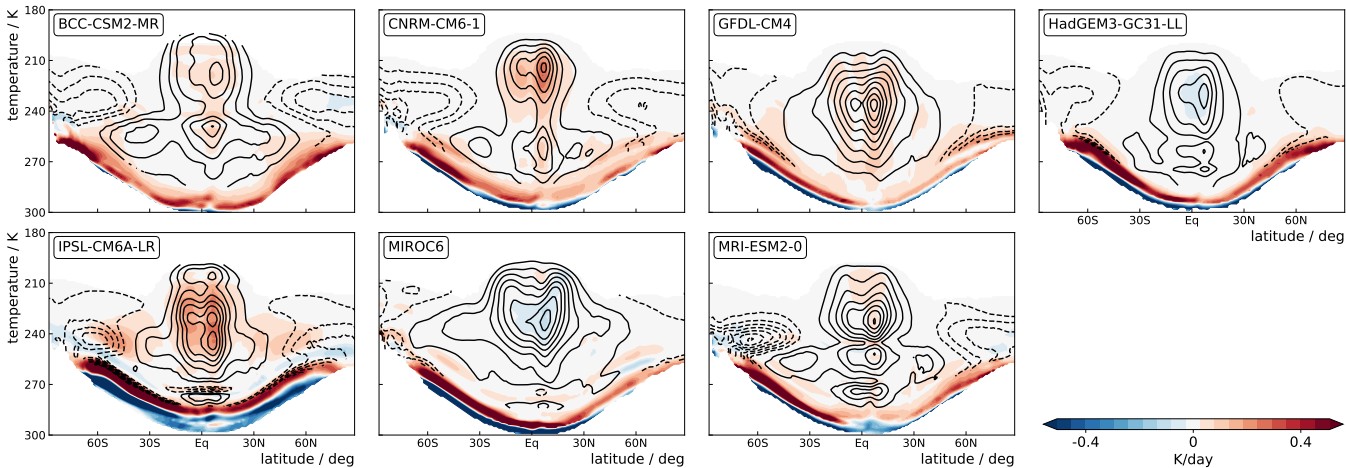

**Figure 8.** Response of cloud-radiative heating to a uniform 4 K ocean surface warming as a function of air temperature. The sampling is limited to the troposphere so that the upper limit of the plotted data marks the tropopause temperature and the lower limit marks the surface temperature. The black lines show the cloud-radiative heating in the amip simulations with a contour spacing of 0.1 K/day. For the latter, only levels with pressures smaller than 750 hPa are taken into account to avoid line cluttering near the surface.

of Jeevanjee and Romps (2018), who combined theory, idealized modeling and global climate model analysis to show that sufficiently away from the surface, all-sky and clear-sky radiative fluxes are independent of climate when measured as a function of temperature. Here, we have extended the view to the radiative heating rates (instead of fluxes) and to clouds (instead of the all-sky or clear-sky atmosphere).

In summary, our comparison of the amip-p4K and amip simulations identifies the upward shift as an excellent prediction of the response of upper-tropospheric cloud-radiative heating to surface warming and shows that the successful prediction results from the strong dependence of clouds and their radiative heating on atmospheric temperature. An important implication is that changes in clouds with warming that go beyond the upward shift are secondary in the global climate models considered and that the overwhelming part of the model differences in the response of cloud-radiative heating to warming is caused by model differences in the simulation of cloud-radiative heating in the present-day climate. This highlights that a prime target for model development should be the improvement of cloud-radiative heating in the present-day climate, for which satellite-based estimates, despite their own uncertainties, can serve as helpful guidelines.

## 5 Response to non-uniform warming of the ocean surface

We now study to what extent the pattern of ocean surface warming affects cloud-radiative heating. To this end, we repeat the analysis of Sec. 4 for the amip-future4K simulations. In these, sea-surface temperature is increased according to a pattern derived from coupled climate models and scaled to an ice-free ocean mean warming of 4 K (Webb et al., 2017). The time-

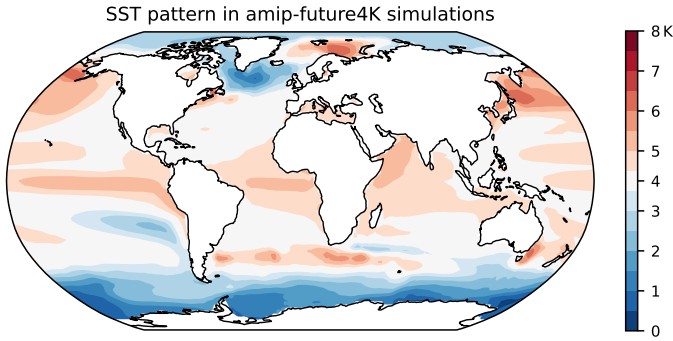

**Figure 9.** Time-mean change in sea-surface temperature (SST) imposed in the amip-future4K simulations. The color map is centered at the global-mean ocean surface warming of 4 K.

mean SST increase is shown in Fig. 9. In the amip-future4K simulations, the tropical ocean warms more than in the amip-4K simulations, in some places by more than 5 instead of the 4 K of the amip-p4K simulations. The ocean warming is muted over the Southern Ocean and the North Atlantic because of ocean heat uptake and changes in the ocean circulation (Armour et al., 2016; Keil et al., 2020). An analysis of simulations with stronger warming patterns would be desirable, for example the CMIP6 CFMIP simulations piSST, piSST-pxK and a4SST (Webb et al., 2017). Yet, only very few models submitted the latter

simulations, and no model included the output needed to diagnose cloud-radiative heating.

    Overall, the results from the amip-future4K simulations closely mirror the results from the amip-p4K simulations. Model differences in cloud-radiative heating are essentially as large in the amip-future4K simulations as in the amip-p4K simulations, and the upper-tropospheric response is very well captured by the same prediction as for the amip-p4K simulations. Because the results are essentially the same, they are not shown in separate figures.

Our main interest here is to show that the pattern of surface warming has little impact on the upper-tropospheric response of cloud-radiative heating, but some impact on the response in the lower troposphere. This is illustrated in Fig. 10, which shows the difference in the cloud-radiative heating between the amip-future4K and amip-p4K simulations. The difference quantifies the extent to which the response of cloud-radiative heating to surface warming depends on the pattern of surface warming. In the upper troposphere, the two sets of simulations agree very well. Thus, sufficiently away from the surface where air

temperatures are mixed by the atmospheric circulation and less strongly tied to spatial variations in surface temperature, cloud-radiative heating is essentially independent of the pattern of surface warming and to first order controlled by the magnitude of global-mean warming. This finding is in line with our previous result that cloud-radiative heating in the upper troposphere is, to very good approximation, a function of atmospheric temperature. The latter implies that cloud-radiative heating is not dependent on how exactly the surface warms, but rather that the surface warming is communicated to the upper troposphere,

where it varies much less spatially. A corollary is that that the upward shift of cloud-radiative heating is as good a predictor for the amip-future4K response as the amip-4K response (using a $\beta$ value of 1.2; not shown).

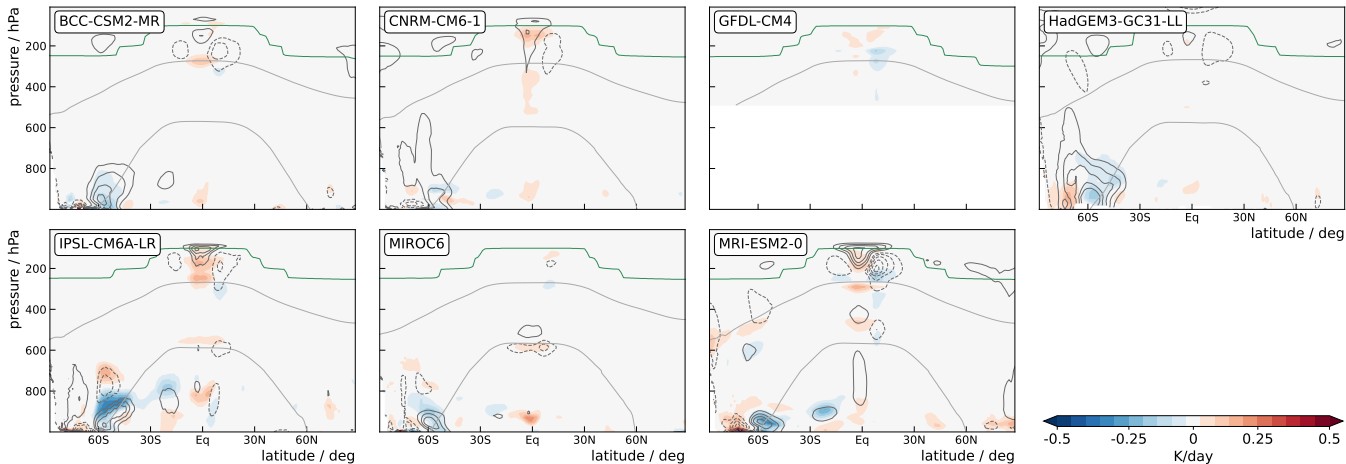

**Figure 10.** Difference between the zonal-mean time-mean cloud-radiative heating in the amip-future4K and amip-p4K simulations (amip-future4K - amip-p4K). The darker gray contour lines show the difference in zonal-mean time-mean cloud fraction between the two simulations (amip-future4K - amip-p4K; solid for positive and dashed for negative values, contour spacing of 0.01). For GFDL-CM4, cloud fraction is not available and cloud-radiative heating is restricted to above 500 hPa because it is derived from zonal-mean heating rates in the amip-future4K simulation but zonally-resolved heating rates in the amip-p4K simulation (cf. Tab. 1). This leads to spurious differences below 500 hPa due to topography. The gray lines mark the 0 and -38 deg C isotherms and the green lines the thermal tropopause for the amip simulations.

We note that low-level clouds and cloud-radiative heating do not follow this paradigm. The response of cloud-radiative heating in the lower troposphere differs between the amip-future4K and amip-p4K simulations, in particular over the Southern Ocean. This is unsurprising since low-level clouds are known to be strongly controlled by sea-surface temperature and lower-tropospheric stability (Wood, 2012; Bretherton, 2015).

In summary, the comparison between the amip-future4K and amip-p4K simulations demonstrates that the response of upper-tropospheric cloud-radiative heating is essentially insensitive to the details of the surface warming. This means that it can be considered a function of the global-mean surface warming, and hence a function of the product of climate sensitivity and radiative forcing.

## 6 Prediction of the response in the upper troposphere from observations

The final step of our analysis is to predict how cloud-radiative heating in the upper troposphere responds to warming. The prediction is independent of the climate models and is obtained solely by combining the physical understanding of cloud-radiative heating and observations. The prediction takes advantage of our findings from global climate models that, first, the response is dominated by an upward shift of the present-day cloud-radiative heating and, second, the pattern of surface warming has little impact on the response.

With these ingredients, the prediction is straightforward to obtain from the approach described in Sect. 4. We shift the present-day cloud-radiative heating upward according to Eq. 6, using $\beta = 1.2$. This is the same value as for the model simulations and corresponds to a global-mean ocean surface warming of 4 K (see Sect. 4). For the present-day cloud-radiative heating, we use the satellite-based estimates from both 2B-FLXHR-LIDAR R05 and CCCM.

Fig. 11 shows the predicted response. Because the prediction is only valid for the upper troposphere, the figure is limited to pressure levels above 600 hPa. Both 2B-FLXHR-LIDAR and CCCM support the arc of anomalous positive cloud-radiative heating in the upper troposphere that extends from the tropics into the high latitudes, consistent with the behavior of the climate models. The arc is more pronounced for CCCM because the present-day cloud-radiative heating in the upper troposphere is stronger and more positive in CCCM than in 2B-FLXHR-LIDAR, as described in Sect. 3. In the tropics and subtropics the prediction is, in a qualitative sense, robust with respect to the satellite product, as is shown by the vertical profiles in the lower panels of Fig. 11. In the extratropics, however, the prediction has a different sign for CCCM and 2B-FLXHR-LIDAR. Thus, the uncertainty in current estimates of cloud-radiative heating precludes a robust prediction in the extratropics.

## 7    Conclusions

We study cloud-radiative heating in an ensemble of 20 CMIP6 models. To this end, we combine model output from four different CMIP6 output tables and, depending on the output available for a given model and simulation, derive cloud-radiative heating either by converting radiative fluxes to heating rates or by directly using the radiative heating rates provided in the CMIP6 archive of the Earth System Grid Foundation. By doing so, we generate the most comprehensive assessment of cloud-radiative heating in global climate models to date, overcoming a limitation of previous work that used only single models (Li et al., 2015; Johansson et al., 2021) or small ensembles of 5 models or less (Cesana et al., 2019; Voigt et al., 2019).

Using simulations of the present-day climate in which sea-surface temperatures are prescribed to observed values, we identify large model differences in cloud-radiative heating in both the lower and upper troposphere. These differences are not unexpected from previous work, which illustrated large model differences in cloud fraction and cloud hydrometeors (Lauer et al., 2023; Li et al., 2016). Differences between models are particularly large in regions where the frozen phase of atmospheric water is prevalent, i.e., in the low-level mixed-phase clouds of the Southern Ocean and the upper-tropospheric ice clouds of the tropics and extratropics. This highlights the challenge of adequately representing cloud ice processes and their interaction with radiation in climate models.

Using simulations in which the ocean surface is warmed, we demonstrate that cloud-radiative heating in the upper troposphere is first and foremost a function of the local air temperature, not surface temperature, and that the response of cloud-radiative heating above 500 hPa to warming is therefore governed by an upward shift of the present-day cloud-radiative heating. This has three important consequences.

1. The first consequence is a new null hypothesis for the response of upper-tropospheric cloud-radiative heating to warming. Because of the tight coupling between clouds and atmospheric temperature in the upper troposphere, the null hypothesis is an upward shift that ensures that cloud-radiative heating is conserved when measured as a function of atmospheric

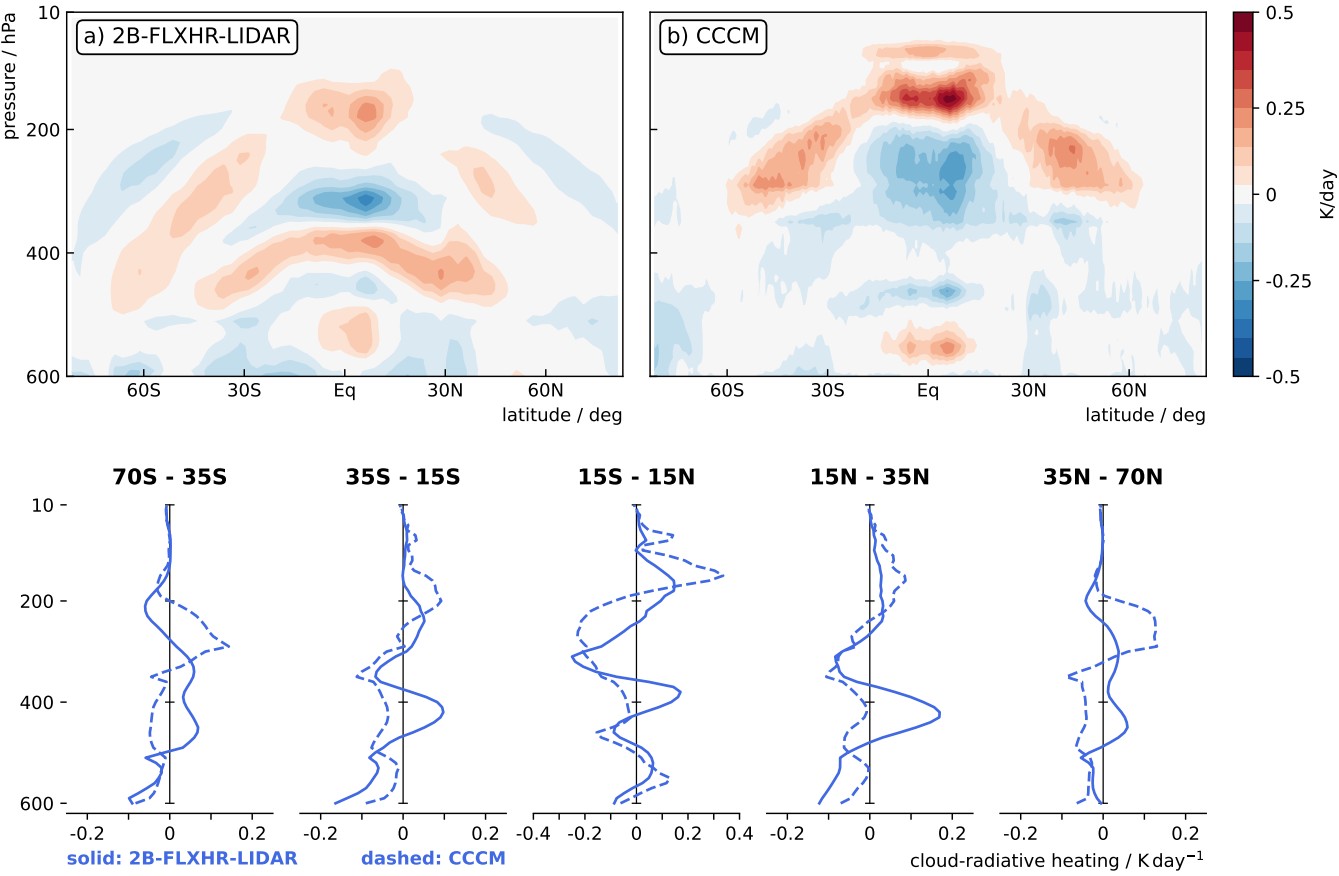

**Figure 11.** Prediction of the response of upper-tropospheric cloud-radiative heating to warming by combining satellite observations from 2B-FLXHR-LIDAR and CCCM with physical understanding. Release R05 of 2B-FLXHR-LIDAR is used. The prediction assumes as global-mean ocean surface warming of 4 K, in line with the amip-p4K and amip-future4K simulations.

temperature. Other changes, such as changes in cloud fraction and cloud ice, are then second order. A corollary of the null hypothesis is that the response of cloud-radiative heating, or more generally clouds, in the upper troposphere should be considered together with, and not separately from, the response of upper-tropospheric temperature. This supports the recent proposal of Yoshimori et al. (2020) for an alternative view on high cloud feedbacks and is analogous to the warming response of water vapor and lapse rates, which are tightly coupled and whose radiative feedbacks on Earth's energy balance are much better understood when considered in combination rather than in isolation (Held and Shell, 2012). The null hypothesis may also help explain why the impact of cloud-radiative changes on the circulation response to warming diagnosed by cloud locking modeling studies can depend on the locked reference state, since cloud locking explicitly breaks the link between clouds and temperature (Albern et al., 2021; Ceppi and Hartmann, 2016; Ceppi and Shepherd, 2017; Huber, 2022).

2. The second consequence is a physics-based prediction. The response of upper-tropospheric cloud-radiative heating is very well predicted by an upward shift of the present-day cloud-radiative heating, with the magnitude of the shift being a function of global-mean surface warming. The prediction is a direct consequence of the null hypothesis described above, and is thus rooted in physical understanding. The prediction captures the upper-tropospheric response independent of the pattern of surface warming because upper-tropospheric temperatures are not strongly sensitive to the details of surface warming thanks due to the homogenizing effect of the atmospheric circulation. This implies that the response of upper-tropospheric cloud-radiative heating can be predicted from the present-day cloud-radiative heating as a function of the magnitude of global-mean surface warming, i.e., it can be predicted by combining knowledge on climate sensitivity and radiative forcing. Notably, the prediction is based entirely on observations of cloud-radiative heating and is thus independent of climate models. However, observational uncertainties limit the prediction currently to the tropics and subtropics. Future work is needed to verify that the prediction is also supported by kilometer-scale climate models, where changes in cloud fraction and cloud hydrometeors might be more pronounced than in the coarse-resolution 50-100 km models used in CMIP6. Future work is also needed to translate the cloud-radiative heating into changes in temperatures and ultimately winds, a task that is non-trivial due to the non-linear nature of the atmospheric circulation and the interaction of radiation with other small-scale diabatic processes.

3. The third consequence is a call to action for model development. Because the present-day cloud-radiative heating provides a tight constraint on the response of cloud-radiative heating to warming in the upper troposphere, future model development efforts should explicitly target cloud-radiative heating. Past efforts have focused on top-of-atmosphere cloud-radiative effects. Despite their uncertainties, satellite observations provide a helpful baseline of the vertical structure of cloud-radiative heating within the atmosphere that has been underutilized for model development and evaluation. Modeling efforts should further include the upcoming km-scale climate models, whose high resolution provides a particular advantage for joint analyses of modeled and observed data, including from upcoming satellite Earth observations such as EarthCare (Illingworth et al., 2015).

Future work should address whether model biases in the simulation of the present-day circulation on climate and weather time scales are related to model differences in cloud-radiative heating. For example, such studies could examine the relationship between the upper-tropospheric tropical cloud-radiative heating and the extratropical jet stream, or the relationship between the extratropical lower- and upper-tropospheric cloud-radiative heating and extratropical cyclones (Voigt et al., 2023). These topics are left for future work.

*Code and data availability.* The software and postprocessed climate model and satellite data is long-term archived in a collection at the Phaidra repository for the permanent secure storage of digital assets at the University of Vienna, https://doi.org/10.25365/phaidra.528. The collection contains: scripts to download CMIP6 model output from ESGF and to analyse and plot CMIP and satellite data (https://doi.org/10.25365/phaidra.528_04); scripts to generate climatologies of cloud-radiative heating from the CloudSat/Calipso 2B-FLXHR-LIDAR satellite product (https://doi.org/10.25365/phaidra.528_03); and postprocessed CMIP6 model output (https://doi.org/10.25365/phaidra.528_01 and

https://doi.org/10.25365/phaidra.528_02). The original CMIP6 model data can be obtained from the Earth System Grid Foundation, and can be downloaded and postprocessed with the scripts given above. Postprocessed zonal-mean time-mean cloud-radiative heating from the 2B-FLXHR-LIDAR and CCCM products is included in https://doi.org/10.25365/phaidra.528_04. The original data for the 2B-FLXHR-LIDAR and CCCM products are available from the CloudSat Data Processing Center (http://www.cloudsat.cira.colostate.edu) and the Atmospheric Science Data Center (https://eosweb.larc.nasa.gov/).


**Appendix A: additional figures**

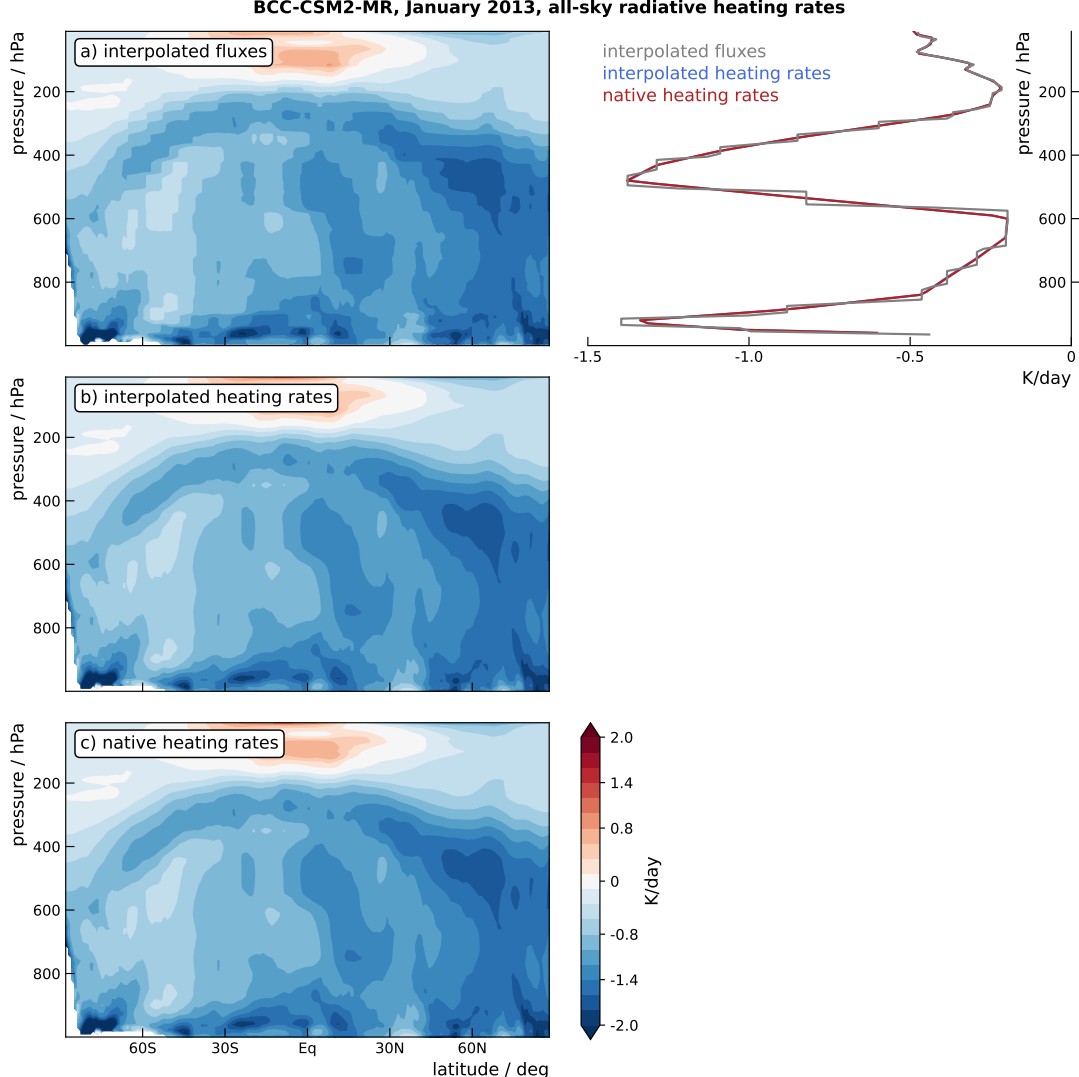

**Figure A1.** Zonal-mean all-sky radiative heating rates in the BCC-CSM2-MR model for January 2013. Panels a and b show radiative heating rates computed from radiative fluxes. For panel a, fluxes are first interpolated from model levels to common pressure levels and heating rates are computed from these interpolated fluxes. For panel b, heating rates are first computed from fluxes on model levels, and these heating rates are then interpolated from model levels to common pressure levels. Throughout the manuscript, we use the order of computation of panel b. Panel c shows the "native" heating rates that were directly output by the model and that we interpolated from model levels to pressure levels. The agreement between panels b and c validates our method to derive heating rates from radiative fluxes (cf. approach 1 of Sect. 2.2), whereas the disagreement between panels a and c illustrates the importance to derive heating rates on model levels. The right panel illustrates this also for a grid point over the Southern Ocean (67 deg S; 68 deg E).

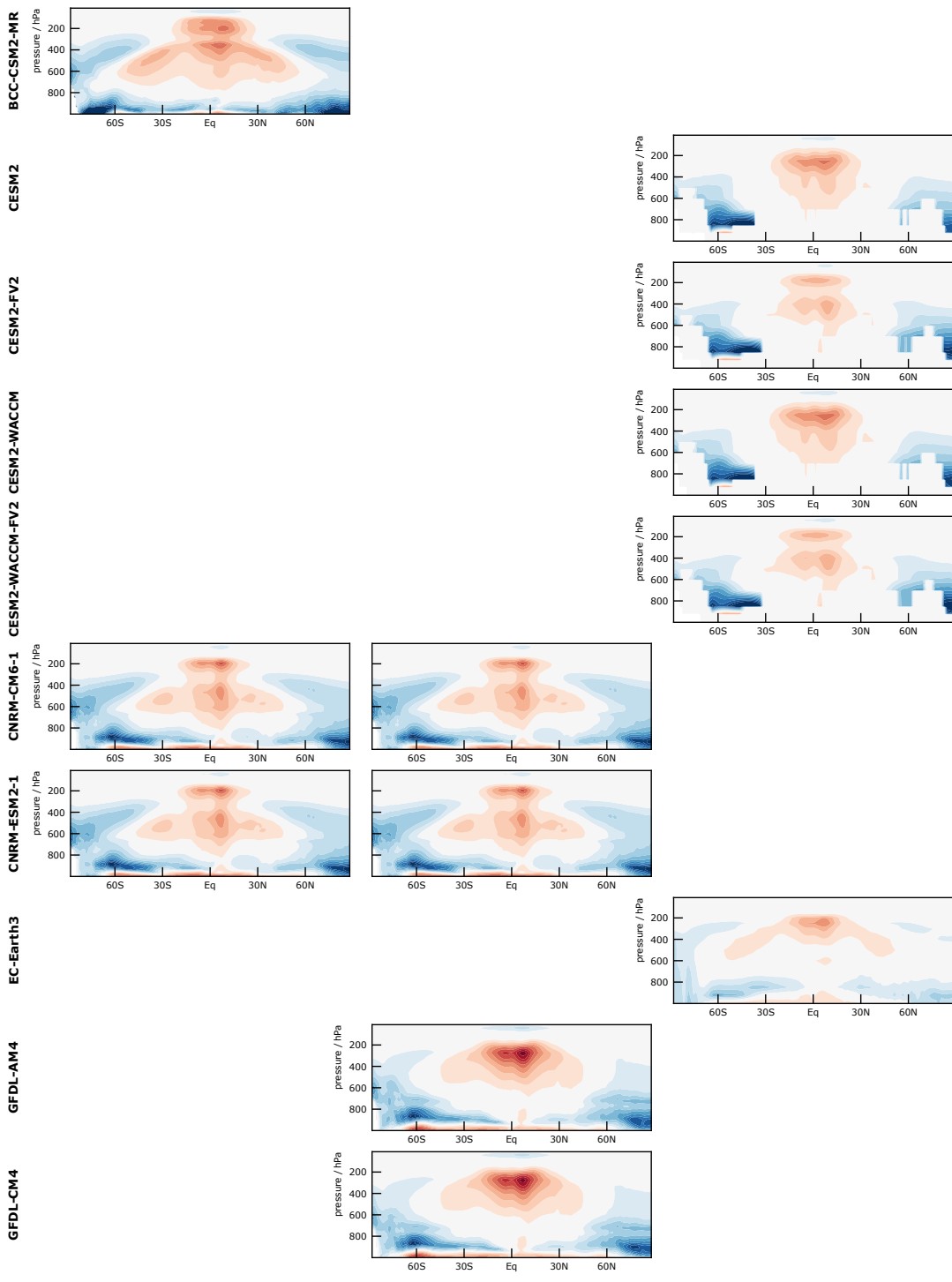

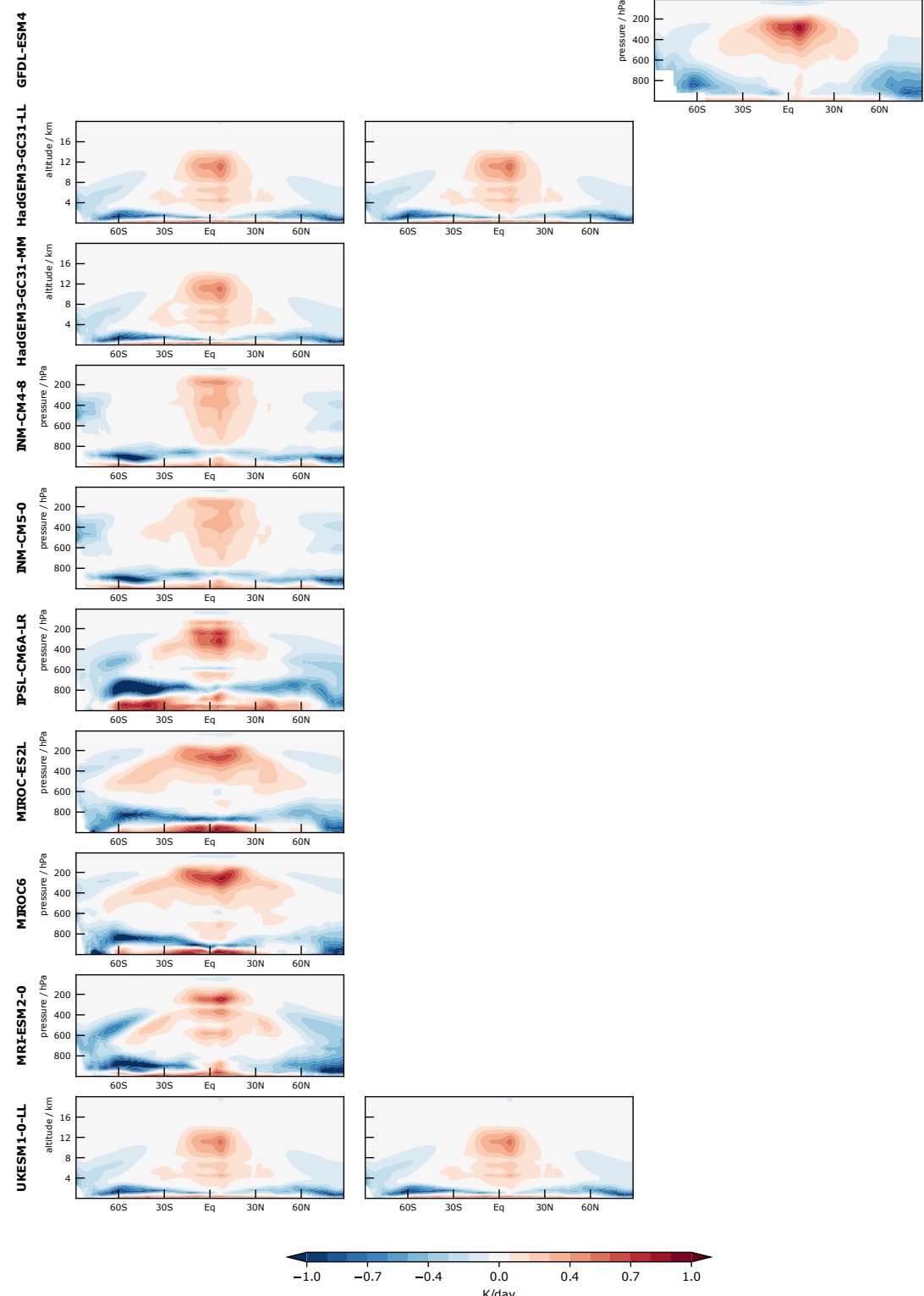

**Figure A2.** Zonal-mean time-mean cloud-radiative heating in amip simulations diagnosed according to the three approaches outlined in Sect. 2 and Tab. 1.

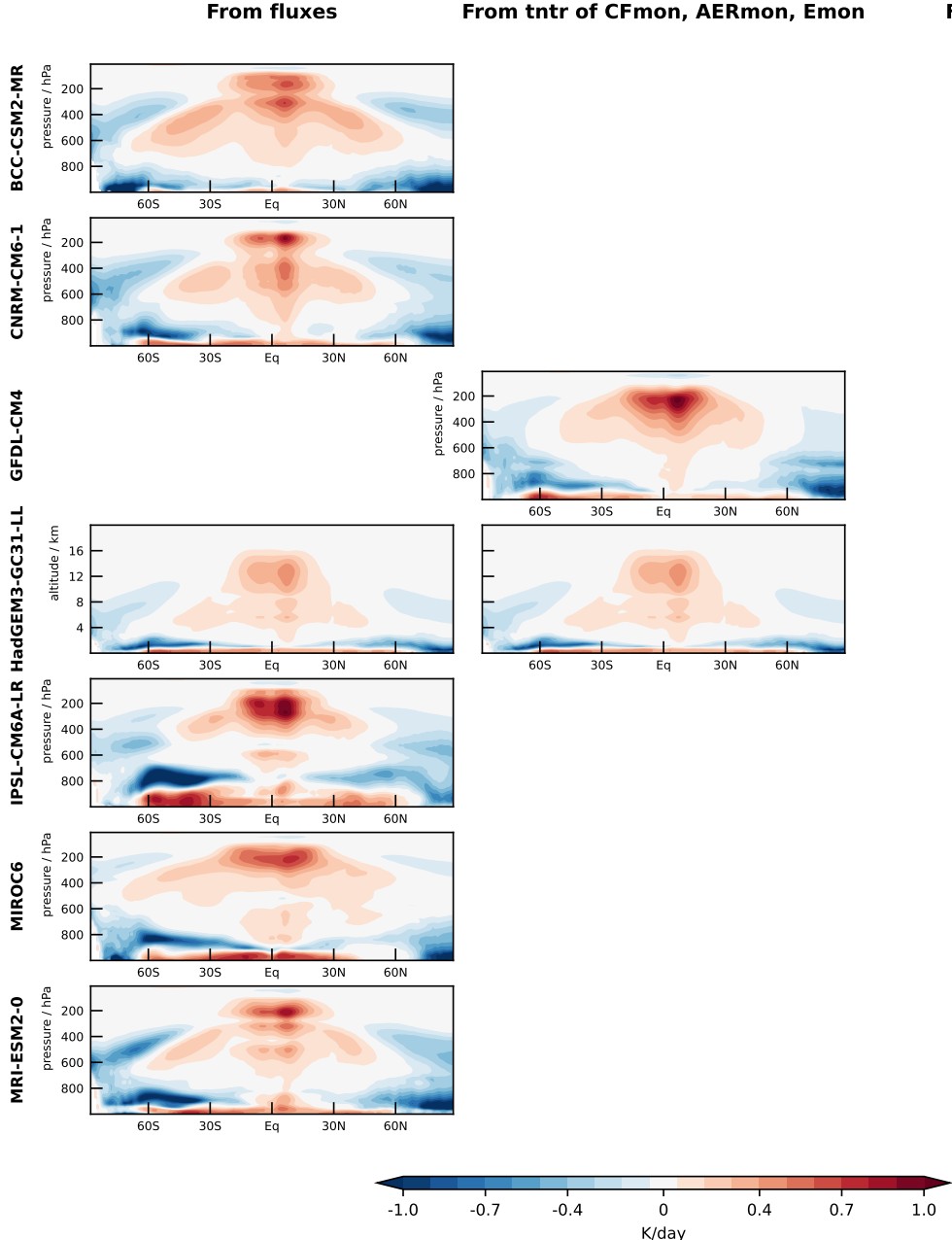

**Figure A3.** Zonal-mean time-mean cloud-radiative heating in amip-p4K simulations diagnosed according to the three approaches outlined in Sect. 2 and Tab. 1.

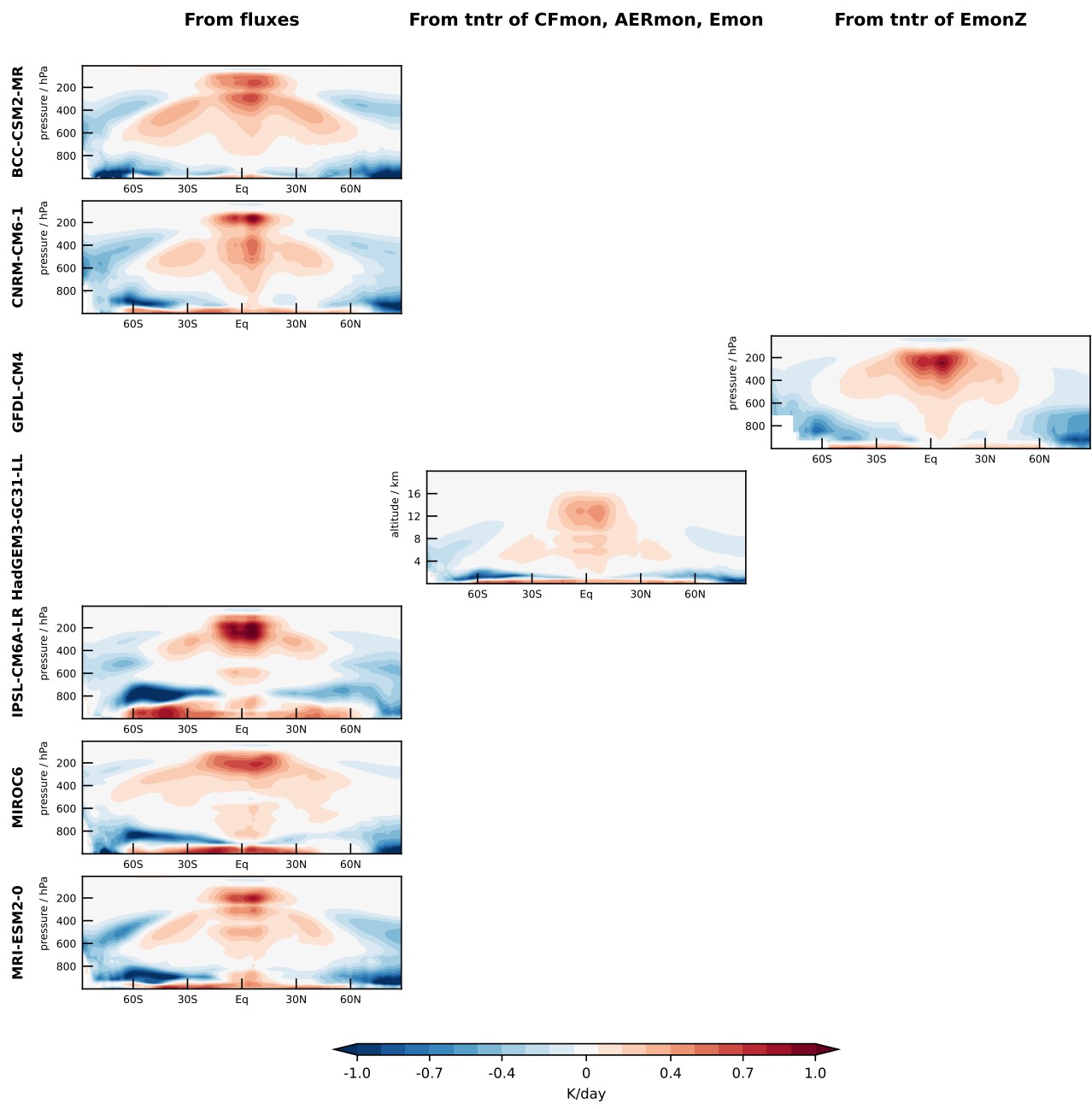

**Figure A4.** Zonal-mean time-mean cloud-radiative heating in amip-future4K simulations diagnosed according to the three approaches outlined in Sect. 2 and Tab. 1.

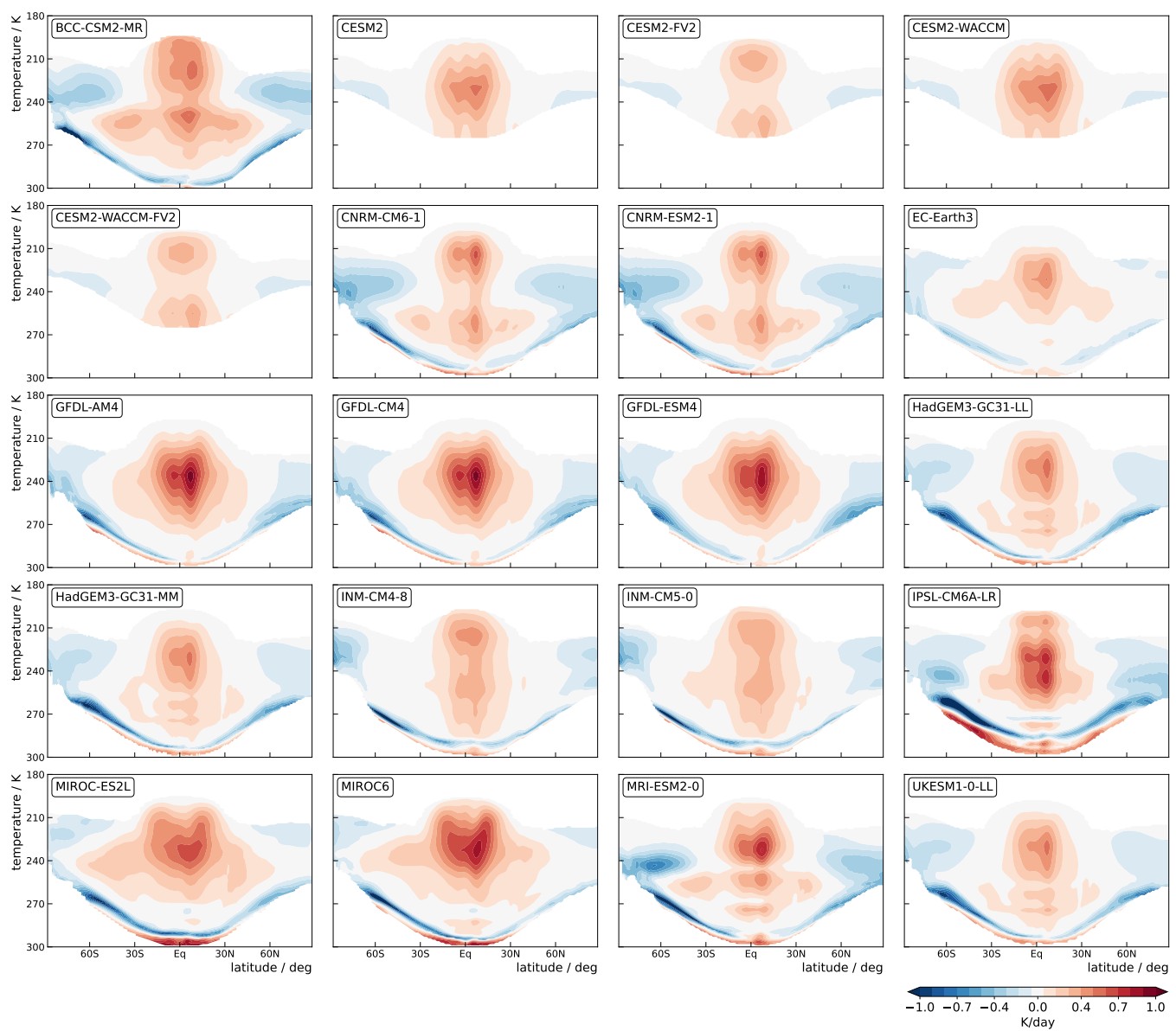

**Figure A5.** Zonal-mean time-mean cloud-radiative heating in amip simulations with 20 CMIP6 models sampled as function of air temperature. The sampling is only done within the troposphere so that the upper limit of the plotted data marks the tropopause temperature and the lower limit marks the surface temperature.

*Author contributions.* The study was designed by AV. Initial exploratory data analysis was performed by SN with input from AV; the data analysis was then extended and finalized by AV. SH calculated the CCCM cloud-radiative heating rates. AV led the writing process of the

paper, with input from all authors. The work is based on the BSc thesis of SN at the Department of Meteorology and Geophysics of the University of Vienna.

*Competing interests.* The contact author has declared that none of the authors has any competing interests.

*Acknowledgements.* We acknowledge the World Climate Research Programme, which, through its Working Group on Coupled Modelling, coordinated and promoted CMIP6. We thank the climate modeling groups for producing and making available their model output, the Earth 445 System Grid Federation (ESGF) for archiving the data and providing access, and the multiple funding agencies who support CMIP6 and ESGF. We thank the developers and maintainers of the open source Python packages NumPy (Harris et al., 2020), Xarray (Hoyer and Hamman, 2017), Matplotlib (Hunter, 2007), MetPy (May et al., 2022), PyTropD (Adam et al., 2018) and CLIMLAB (Rose, 2018). Writing was assisted by the AI tool DeepL Write in terms of grammar and wording. Open acccess funding provided by the University of Vienna.

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
