# Peer review of "Atmospheric cloud-radiative heating in CMIP6 and observations, and its response to surface warming"

_EGUsphere, 2023_

## Author Comment (AC1)

**Atmospheric cloud-radiative heating in CMIP6 and observations, and its response to surface warming - Response to Reviewers**

**Aiko Voigt, Stefanie North, Blaz Gasparini, and Seung-Hee Ham**

We thank the reviewers for their thoughtful evaluation and constructive comments. Below, we describe how we plan to revise the manuscript. We are confident that this will improve the manuscript, and are optimistic that the revised manuscript will satisfy the high standards of ACP. The reviewers' comments are in black font, our answers are in blue font. Note that we abbreviate atmospheric cloud-radiative heating as CRH in the following.

**Reviewer 1 (https://doi.org/10.5194/egusphere-2023-2612-RC1)**

**Summary**

The authors perform a systematic assessment of in-atmosphere cloud radiative effects across CMIP models, and compare these with two observationally-based estimates. They also examine the climate change response of the cloud radiative heating in models and show that the upper tropospheric response is well modeled by the assumption that clouds shift upwards following the isotherms, consistent with theoretical constraints in the literature. The results are interesting and important, and the paper is well written. I have only minor comments and recommend acceptance after they are considered.

**General Comments**

1. The paper occasionally uses language that seems a bit self-promoting, which kind of grates on me for a paper (it might be appropriate in a proposal). For example, I note the following phrases describing the work: "the first systematic assessment"; "the most comprehensive assessment ever generated"; "the most comprehensive assessment . . . to date"; "is a major step forward"; "we generate the most comprehensive assessment of cloud radiative heating in global climate models to date". Suggest using this language sparingly.

Agreed. While we find it important to clearly communicate that our manuscript provides the first assessment of atmospheric cloud-radiative heating across more than a handful of models, we agree that the tone should be adapted. We will revise the text accordingly.

2. Can the mean state cloud properties and the upward shift of clouds be separated as is done here or are they inextricably linked? By this I mean, if a biased model had the "right" mean-state cloud radiative heating profile, would it still shift upwards by the same amount as it currently does, or would that corrected mean-state cloud radiative heating then lead to a different response of either the temperature profile or the circulation influenced by the diabatic heating? If not, does the fact that the warming-induced high cloud response can be modeled as a simple upward shift somewhat undercut the message that cloud radiative heating is super important for the atmospheric circulation and its response to warming (used as motivation for the analysis).

Thanks for this interesting thought. Since the upward shift works well across models, which differ strongly in their present-day CRH, the upward shift itself does not appear to be tied to the ability of a model to capture the "right" present-day CRH. Also, as stated already in one of our three main points in the conclusions, future work should use the upward shift as a null-hypothesis for the high-level CRH response to warming and work out deviations from this that could arise from changes in ice water content or cloud fraction.

3. The "pattern effect" results: While the amip-future4K experiment does impose a pattern that is not uniform, it is not a strongly heterogeneous pattern and typically most pattern effect studies contrast the uniform or 4xCO2 warming pattern with something more distinct, like the observed warming over the last few decades. Hence while I don't doubt that the high cloud response is going to be pretty similar even with a more heterogeneous warming pattern, it is too strong a statement to state unequivocally that the difference between these two simulations "quantifies the extent to which the response of cloud-radiative heating to surface warming depends on the pattern of surface warming" and to conclude that "the pattern of surface warming has little impact", that "cloud-radiative heating is essentially independent of the pattern of surface warming", or that "the response of upper tropospheric cloud-radiative heating is essentially insensitive to the details of the surface warming." All that has been tested here is whether the high cloud response is different between amip-p4K and amip-future4K, which is a weak litmus test given how similar these patterns are. I suggest either weakening / appropriately caveating these statements; contrasting the cloud heating response in experiments with more distinct warming patterns; or just dropping this part of the analysis, which seems a bit tangential anyway.

We agree that simulations with an even stronger SST pattern would be fantastic. We have looked into possibilities to quantify the pattern effect on CRH also from other simulations. To do so, one requires three simulations: a control simulation, a warming simulation, and a warming simulation with the same level of global-mean warming but a different pattern of surface warming. The CMIP6 CFMIP simulations piSST, piSST-pxK and a4SST (Webb et al., 2017) provide these, yet the CMIP6 ESGF archive provides such simulations only for a few models, and no model includes the output needed to diagnose CRH.

That being said, the surface warming pattern in the amip-future4K simulations is substantial in our opinion, and strong in some regions (e.g., the Southern Ocean; see Fig. 7 of the submitted version). We also consider the pattern analysis an important part of our manuscript: it indicates that in order to predict the warming response of high-level clouds and their CRH, knowledge of the global-mean surface warming is sufficient. Together with the upward shift based on the $\beta$ parameter, this opens the door to incorporate the high-level CRH response into a quantitative theory of global-mean warming much more easily than the low-level cloud response, which more strongly depends on the pattern of surface warming. Still, we will revise the text to clarify that simulations with a stronger warming pattern would be helpful.

Please note that we have discovered an error in the colorbar of Fig. 7. The colorbar needs to be centered at 4K, which is the global mean SST increase, and range from 0K to 8K. The colorbar will be corrected in the revised manuscript.

**Specific Comments**

L83: Unequivocally is misspelled.

Thanks, this will be corrected.

Figure 1 description: I suggest dedicating some more text to explaining the basic features of this figure (or Figure 2) for those readers that are not used to looking at in-atmosphere cloud radiative heating rates.

Will do. We will also also include a subplot for observed zime-mean zonal-mean cloud cover in Fig. 2 to illustrate the link between clouds and their radiative heating. The cloud cover will be taken from Bertrand et al. (2024).

I am surprised that the same Beta parameter works for every model in Eq. 5. Is this because they are all subject to the same uniform+4K of SST warming and have roughly the same upward shift of isotherms? If

one were to estimate the "best fit" Beta for each model, how much would it vary, and would that lead to even better predictions? This connects back to General Comment #2 where it seems that the basic atmospheric response is not very dependent on cloud radiative heating such that one can model each response using a single model-invariant Beta value.

As derived and described in detail by Singh and O'Gorman (2012), $\beta$ is proportional to the warming at the boundary layer top (see their Eq. 15). We assume a boundary layer top warming of 5 K, which leads to $\beta = 1.2$. Because all models use the same +4K surface warming, we use the same $\beta$ value. We also use the same value for the amip-future4K simulations with patterned SST warming.

Thanks for the suggestion to fit $\beta$ individually for each model. Yet, our primary aim is to show that the high-level CRH change can be predicted based from the magnitude of surface warming and the present-day CRH. This point is well made by using the same value for each model. We hence decided against fitting an optimal value of $\beta$ for each model.

We found an error on L223 of the original submission. Eq. 12 should read Eq. 15. This will be corrected.

Figure 4: suggest showing the multi-model mean or median in the last open panel. Rather than overlaying the isotherms, I wonder if it might be helpful to instead overlay the amip control climate cloud radiative heating contours.

Given the stark differences in the CRH response between models, we see little added value in plotting the model median. However, we will experiment with overlaying the amip control CRH.

Figure 7: Suggest noting in the caption rather than in the text that the colormap is centered on 4K.

Good idea, will do.

L345: In this discussion, I suggest citing Yoshimori et al (2020) [DOI: 10.1175/JCLI-D-19-0108.1], who make many of these points.

Thanks for pointing us to the interesting work of Yoshimori et al. We will cite the work in the discussion section.

**Reviewer 2 (https://doi.org/10.5194/egusphere-2023-2612-RC2)**

This paper examines the cloud radiative effect from an angle different from many other studies. Instead of measuring the cloud effect by the radiative fluxes, the analysis here is focused on the radiative heating of the atmosphere. This relatively new angle presents many interesting results and mainly for this reason, I think the paper potentially makes a valuable contribution to our understanding of the climate effects of clouds. I would recommend acceptance if the following comments were addressed.

**Method**

It is well recognized that the cloud radiative effect, if simply measured by clear- and all-sky difference (eq. 1), would be subject to large biases due to the non-cloud changes causing different radiative changes in clear- and all-sky (a "masking" effect). The kernel-based adjustment method of Shell et al. (2008, https://doi.org/10.1175/2007JCLI2044.1), for example, is widely adopted to correct this bias in the cloud feedback analysis. It can be expected that similar issues will occur for the heating rate analysis. What measure can/should be applied here? Tests, discussions, and/or recommendations should be made, in the context of continued kernel developments. For example, one latest kernel dataset published by Huang and Huang

[Figure]

Figure R1: Variance across amip model simulations for zonal-mean time-mean all-sky radiative heating. The variance in all-sky heating is decomposed into variances for clear-sky and cloud-radiative heating and the covariance between the latter two. By design, the sum of the variances in clear-sky radiative heating and cloud-radiative heating and the covariance add up to the all-sky variance.

2023, https://doi.org/10.5194/essd-15-3001-2023 extended the kernels from TOA to surface already. Would you recommend making available layerwise kernels for heating rates or (equivalently) for flux profiles?

Our recommendation is to diagnose CRH from native model output, instead of using kernels. This is because radiation is a non-linear function of the atmospheric state, and because to convert radiative fluxes to heating rates in a feasible manner requires to compute radiative flux divergence on model levels (see also our answer to the reviewer's following comment).

As for masking effects, we have computed all-sky, clear-sky and cloud-radiative heating for all amip simulations. Based on this, we have further computed the variance of all-sky heating and decomposed this into the sum of variances of clear-sky and cloud-radiative heating and the covariance of clear-sky and cloud-radiative heating. This analysis is illustrated in Fig. R1 and shows that overall, model spread in all-sky radiative heating is dominated by model spread in cloud-radiative heating. The figure and a corresponding paragraph will be included in the revised manucript.

Line 149. Given that CRH profile is discrete, I am concerned about the impacts of the interpolation on the resulted profiles. For example, are the CRH features at the cloud boundaries, e.g., at the top of high

clouds and around boundary layer clouds, dislocated or blurred by this processing? Some discussions, preferably with supporting plots, would be appreciated to clear this concern. This could for example affect the replotting of the CRH results in different coordinates later on.

We agree that this is an important point to consider, and we have considered it in our analysis already. For models for which radiative heating rates are computed from radiative fluxes, we first compute heating rates on model levels and then interpolate the latter to pressure levels. Fig. R2 illustrates the importance of this, and shows that large errors can occur when radiative fluxes are first interpolated to pressure levels and radiative heating rates are computed from interpolated fluxes. Specifically, the figure shows this for the all-sky radiative heating rate for the amip simulation of BCC-CSM2-MR, for which also the online diagnosed radiative heating is available. The comparison between the latter and our derived radiative heating shows that our diagnostic method works. The figure and a corresponding paragraph will be included in the revised manuscript.

**Results**

The paper is claimed to be "the most comprehensive assessment of atmospheric cloud radiative heating" (line 48, 324). I found the lack of such crucial results as the longwave vs. shortwave decomposition of the heating rates, at odds with the claim. These may be necessary for interpreting some results, e.g., the cancellation noted in Line 239.

Our paper is the most comprehensive assessment because it for the first time assesses CRH across more than a handful of model. We will clarify this. Follow-up studies might look into more fine-grained aspects of CRH, such as its shortwave and longwave components, yet we prefer to not include such analyses to limit the length of the manuscript (which is already quite long in our opinion) and keep a clear focus. We will mention possibilities for follow-up work in the conclusions of the revised manuscript, however.

The cancellation mentioned by the reviewer arises because the CRH pattern is not parallel to pressure surfaces, not because of a cancellation between shortwave and longwave heating. The "slanted" pattern of CRH in the extratropics indeed is one reason for sampling CRH in terms of air temperature.

Line 170: warming near surface seems not as noticeable from Li et al. (https://doi.org/10.1175/JCLI-D-14-00825.1)?

We are not sure what the reviewers refers to. Li et al. is based on a single model (IPSL-CM5A-LR), while we are looking at 20 models. In the model studied in Li et al., the positive CRH near the surface is not strongly visible (see their Fig. 4). We suspect this is because Li et al. use a non-linear pressure axis that squashes the near-surface atmosphere.

Line 177: Is this difference due to cloud difference or other variables? Perhaps good to overlay these plots with cloud amount climatology.

This is difficult to say, and could be due to difference in cloud cover, cloud ice and cloud liquid content, and radiative treatment of clouds (e.g., optical properties of ice crystals). That being said, we cannot trace the sources of the model differences here. Nevertheless, we will add a plot in the appendix for zonal-mean time-mean cloud cover in the amip simulations.

Line 188: what's causing their difference, given they're based on the same active sensors? Any systematic bias in obs, e.g., due to their limited time/space sampling, when compared to GCMs?

The difference in the two observation-based estimates of CRH from 2B-FLXHR-LIDAR and CCCM is

[Figure]

Figure R2: All-sky radiative heating for the amip simulation with BCC-CSM2-MR. a) heating rate diagnosed when radiative fluxes are first interpolated to pressure levels. b) heating rate diagnosed when the heating rate is computed from fluxes on model levels and then interpolated to pressure levels. c) the native heating rate provided as model output on model levels and interpolated to pressure levels. The left plot shows a gridbox in the Southern Ocean.

indeed somewhat sobering. The reasons for these differences have been studied in detail by Ham et al. (2017). We will refer to this work in the revision and summarize its main points.

Line 227: beta seems a crucial parameter in this analysis. More explanation and discussion on how this value is set would be helpful.

Agreed. We will add more background on $\beta$ and the choice of its value. See also our answer to a similar comment by reviewer 1.

Line 247: "very similar" sounds subjective to me.

In 5 of 7 models, the CRH response is less than 0.1 K/day when sampled as a function of atmospheric temperature and apart from the boundary layer in Fig. 6. Compared to Fig. 4, which samples the response as a function of pressure, this is indeed "very similar" in our view. We acknowledge that this is a subjective statement, but is not any assessment subjective at some level? Still, we will substantiate our statement by including that the difference is less than 0.1 K/day for most models.

Line 292/298: there seems latitudinal difference which may be a different aspect of Ts control?

We are not sure what the reviewer refers to.

**Literature review**

The introduction and comparison of results to previous works would benefit from a more complete inclusion of relevant papers, such as: Zhang et al. (2017), https://doi.org/10.1007/s00382-016-3501-0 and Kato et al. (2019), https://doi.org/10.1029/2018JD028878.

Thanks, we will add these papers. We also learned about the recent work of Luo et al. (2023), which we will add as well.

**References**

Bertrand, L., J. E. Kay, J. Haynes, and G. de Boer, 2024: A global gridded dataset for cloud vertical structure from combined CloudSat and CALIPSO observations. Earth Syst. Sci. Data, 16 (3), 1301–1316, doi:10.5194/essd-16-1301-2024.

Ham, S.-H., et al., 2017: Cloud occurrences and cloud radiative effects (CREs) from CERES-CALIPSO-CloudSat-MODIS (CCCM) and CloudSat radar-lidar (RL) products. J. Geophys. Res. Atmos., 122 (16), 8852–8884, doi:10.1002/2017JD026725.

Luo, H., J. Quaas, and Y. Han, 2023: Examining cloud vertical structure and radiative effects from satellite retrievals and evaluation of CMIP6 scenarios. Atmos. Chem. Phys., 23 (14), 8169–8186, doi:10.5194/acp-23-8169-2023.

Singh, M. S. and P. A. O'Gorman, 2012: Upward Shift of the Atmospheric General Circulation under Global Warming: Theory and Simulations. J. Climate, 25 (23), 8259–8276, doi:10.1175/JCLI-D-11-00699.1.

Webb, M. J., et al., 2017: The Cloud Feedback Model Intercomparison Project (CFMIP) contribution to CMIP6. Geosci. Model Dev., 10 (1), 359–384, doi:10.5194/gmd-10-359-2017.

---

## Author Response (AR1)

**Atmospheric cloud-radiative heating in CMIP6 and observations, and its response to surface warming - Response to Reviewers**

**Aiko Voigt, Stefanie North, Blaz Gasparini, and Seung-Hee Ham**

We thank the reviewers for their thoughtful evaluation and constructive comments. Below, we describe how we have revised the manuscript, following our plans articulated in our response as part of the discussion phase. We hope that the revised manuscript will satisfy the reviewers' comments and the high standards of ACP.

Throughout the document, the reviewers' comments are in black font, our answers are in blue font. Note that we abbreviate atmospheric cloud-radiative heating as CRH in the following. Besides the changes described here, we have made editorial changes to improve the readability of the manuscript. These changes do not alter the meaning or content and can be traced in the tracked-changes version.

**Reviewer 1 (https://doi.org/10.5194/egusphere-2023-2612-RC1)**

**Summary**

The authors perform a systematic assessment of in-atmosphere cloud radiative effects across CMIP models, and compare these with two observationally-based estimates. They also examine the climate change response of the cloud radiative heating in models and show that the upper tropospheric response is well modeled by the assumption that clouds shift upwards following the isotherms, consistent with theoretical constraints in the literature. The results are interesting and important, and the paper is well written. I have only minor comments and recommend acceptance after they are considered.

**General Comments**

1. The paper occasionally uses language that seems a bit self-promoting, which kind of grates on me for a paper (it might be appropriate in a proposal). For example, I note the following phrases describing the work: "the first systematic assessment"; "the most comprehensive assessment ever generated"; "the most comprehensive assessment . . . to date"; "is a major step forward"; "we generate the most comprehensive assessment of cloud radiative heating in global climate models to date". Suggest using this language sparingly.

Agreed. While we find it important to clearly communicate that our manuscript provides the first assessment of atmospheric cloud-radiative heating across more than a handful of models, we agree that the tone needed to be adapted. According changes have been made throughout the text. See, e.g., L5, L52-55 and L394-395 in the tracked-changes version.

2. Can the mean state cloud properties and the upward shift of clouds be separated as is done here or are they inextricably linked? By this I mean, if a biased model had the "right" mean-state cloud radiative heating profile, would it still shift upwards by the same amount as it currently does, or would that corrected mean-state cloud radiative heating then lead to a different response of either the temperature profile or the circulation influenced by the diabatic heating? If not, does the fact that the warming-induced high cloud response can be modeled as a simple upward shift somewhat undercut the message that cloud radiative heating is super important for the atmospheric circulation and its response to warming (used as motivation for the analysis).

Thanks for this interesting thought. Since the upward shift works well across models, which differ strongly

in their present-day CRH, the upward shift itself does not appear to be tied to the ability of a model to capture the "right" present-day CRH. Also, as stated already in one of our three main points in the conclusions, future work should use the upward shift as a null-hypothesis for the high-level CRH response to warming and work out deviations from this that could arise from changes in ice water content or cloud fraction.

3. The "pattern effect" results: While the amip-future4K experiment does impose a pattern that is not uniform, it is not a strongly heterogeneous pattern and typically most pattern effect studies contrast the uniform or 4xCO2 warming pattern with something more distinct, like the observed warming over the last few decades. Hence while I don't doubt that the high cloud response is going to be pretty similar even with a more heterogeneous warming pattern, it is too strong a statement to state unequivocally that the difference between these two simulations "quantifies the extent to which the response of cloud-radiative heating to surface warming depends on the pattern of surface warming" and to conclude that "the pattern of surface warming has little impact", that "cloud-radiative heating is essentially independent of the pattern of surface warming", or that "the response of upper tropospheric cloud-radiative heating is essentially insensitive to the details of the surface warming." All that has been tested here is whether the high cloud response is different between amip-p4K and amip-future4K, which is a weak litmus test given how similar these patterns are. I suggest either weakening / appropriately caveating these statements; contrasting the cloud heating response in experiments with more distinct warming patterns; or just dropping this part of the analysis, which seems a bit tangential anyway.

We agree that simulations with an even stronger SST pattern would be fantastic. We have looked into possibilities to quantify the pattern effect on CRH also from other simulations. To do so, one requires three simulations: a control simulation, a warming simulation, and a warming simulation with the same level of global-mean warming but a different pattern of surface warming. The CMIP6 CFMIP simulations piSST, piSST-pxK and a4SST (Webb et al., 2017) provide these, yet the CMIP6 ESGF archive provides such simulations only for a few models, and no model includes the output needed to diagnose CRH.

That being said, the surface warming pattern in the amip-future4K simulations is substantial in our opinion, and strong in some regions (e.g., the Southern Ocean; see Fig. 7 of the submitted version). We also consider the pattern analysis an important part of our manuscript: it indicates that in order to predict the warming response of high-level clouds and their CRH, knowledge of the global-mean surface warming is sufficient. Together with the upward shift based on the $\beta$ parameter, this opens the door to incorporate the high-level CRH response into a quantitative theory of global-mean warming much more easily than the low-level cloud response, which more strongly depends on the pattern of surface warming. Still, we have revised the text to clarify that simulations with a stronger warming pattern would be helpful for future studies (see L336-339 of the tracked-changes version).

Please note that we have discovered an error in the colorbar of Fig. 7. The colorbar needs to be centered at 4K, which is the global mean SST increase, and range from 0K to 8K. This has been corrected (please note that the figure is now Fig. 8).

**Specific Comments**

L83: Unequivocally is misspelled.

Thanks, corrected.

Figure 1 description: I suggest dedicating some more text to explaining the basic features of this figure (or Figure 2) for those readers that are not used to looking at in-atmosphere cloud radiative heating rates.

Thanks for the suggestion. We have added a desription of the basic features of cloud-radiative heating and their connection to the cloud pattern. We have further added the new Fig. 2 that shows cloud fraction in the amip simulations, and we have added observed cloud fraction in Fig. 3 from Bertrand et al. (2024). Please see the new Subsect. 2.4 and L200-206 of the tracked-changes version.

I am surprised that the same Beta parameter works for every model in Eq. 5. Is this because they are all subject to the same uniform+4K of SST warming and have roughly the same upward shift of isotherms? If one were to estimate the "best fit" Beta for each model, how much would it vary, and would that lead to even better predictions? This connects back to General Comment #2 where it seems that the basic atmospheric response is not very dependent on cloud radiative heating such that one can model each response using a single model-invariant Beta value.

As derived and described in detail by Singh and O'Gorman (2012), $\beta$ is proportional to the warming at the boundary layer top (see their Eq. 15). We have calculated the model-mean warming at 800 hPa in the amip-p4K simulations as 4.8 K, which rounded to 5 K yields $\beta = 1.2$. Because all models use the same +4K surface warming, the model spread in the warming at 800 hPa is below 0.3 K, justifying the use of the same $\beta$ value for all models. This is now explained in L280-284 of the tracked-changes version.

Thanks for the suggestion to fit $\beta$ individually for each model. Yet, our primary aim is to show that the high-level CRH change can be predicted based from the magnitude of surface warming and the present-day CRH. This point is well made by using the same value for each model. We hence decided against fitting an optimal value of $\beta$ for each model. This thinking is made explicit in L284-285 of the tracked-changes version.

Figure 4: suggest showing the multi-model mean or median in the last open panel. Rather than overlaying the isotherms, I wonder if it might be helpful to instead overlay the amip control climate cloud radiative heating contours.

Given the stark differences in the CRH response between models, we see little added value in plotting the model median. We have experimented with overlaying the amip control CRH but found this to not work well visually. However, we have removed the isotherms for the amip-p4K simulations.

Figure 7: Suggest noting in the caption rather than in the text that the colormap is centered on 4K.

Done, thanks. See Fig. 9 of the revised manuscript.

L345: In this discussion, I suggest citing Yoshimori et al (2020) [DOI: 10.1175/JCLI-D-19-0108.1], who make many of these points.

Thanks for pointing us to the interesting work of Yoshimori et al., which we now cite in the discussion section. See L407-408 of the tracked-changes version.

**Reviewer 2 (https://doi.org/10.5194/egusphere-2023-2612-RC2)**

This paper examines the cloud radiative effect from an angle different from many other studies. Instead of measuring the cloud effect by the radiative fluxes, the analysis here is focused on the radiative heating of the atmosphere. This relatively new angle presents many interesting results and mainly for this reason, I think the paper potentially makes a valuable contribution to our understanding of the climate effects of clouds. I would recommend acceptance if the following comments were addressed.

**Method**

It is well recognized that the cloud radiative effect, if simply measured by clear- and all-sky difference (eq. 1), would be subject to large biases due to the non-cloud changes causing different radiative changes in clear- and all-sky (a "masking" effect). The kernel-based adjustment method of Shell et al. (2008, https://doi.org/10.1175/2007JCLI2044.1), for example, is widely adopted to correct this bias in the cloud feedback analysis. It can be expected that similar issues will occur for the heating rate analysis. What measure can/should be applied here? Tests, discussions, and/or recommendations should be made, in the context of continued kernel developments. For example, one latest kernel dataset published by Huang and Huang 2023, https://doi.org/10.5194/essd-15-3001-2023 extended the kernels from TOA to surface already. Would you recommend making available layerwise kernels for heating rates or (equivalently) for flux profiles?

Our recommendation is to diagnose CRH from native model output, instead of using kernels. This is because radiation is a non-linear function of the atmospheric state, and because to convert radiative fluxes to heating rates in a feasible manner requires to compute radiative flux divergence on model levels (see also our answer to the reviewer's following comment).

As for masking effects, we have computed all-sky, clear-sky and cloud-radiative heating for all amip simulations. Based on this, we have further computed the variance of all-sky heating and decomposed this into the sum of variances of clear-sky and cloud-radiative heating and the covariance of clear-sky and cloud-radiative heating. This analysis is illustrated in the new Fig. 4 and shows that overall, model spread in all-sky radiative heating is dominated by model spread in cloud-radiative heating. We have also added a new paragraph on this, see L224-231 of the tracked-changes version.

Line 149. Given that CRH profile is discrete, I am concerned about the impacts of the interpolation on the resulted profiles. For example, are the CRH features at the cloud boundaries, e.g., at the top of high clouds and around boundary layer clouds, dislocated or blurred by this processing? Some discussions, preferably with supporting plots, would be appreciated to clear this concern. This could for example affect the replotting of the CRH results in different coordinates later on.

We agree that this is an important point to consider, and we have considered it in our analysis already. For models for which radiative heating rates are computed from radiative fluxes, we first compute heating rates on model levels and then interpolate the latter to pressure levels. This is illustrated in the new additional figure A1, and described in the text. See L144-146 of the tracked-changes version and the caption of new figure A1.

**Results**

The paper is claimed to be "the most comprehensive assessment of atmospheric cloud radiative heating" (line 48, 324). I found the lack of such crucial results as the longwave vs. shortwave decomposition of the heating rates, at odds with the claim. These may be necessary for interpreting some results, e.g., the cancellation noted in Line 239.

Our paper is the most comprehensive assessment because it for the first time assesses CRH across more than a handful of model. This is pointed out even more clearly in the revised manuscript, please see L52-61 of the tracked-changes version. Following a related remark on language by reviewer 1, we have also adapted the tone at several places. See, e.g., L5, L52-55 and L394-395 in the tracked-changes version. We agree, however, that follow-up studies might look into more fine-grained aspects of CRH, such as its shortwave and longwave components, yet we prefer to not include such analyses to limit the length of the manuscript

(which is already quite long in our opinion) and keep a clear focus.

The cancellation mentioned by the reviewer arises because the CRH pattern is not parallel to pressure surfaces, not because of a cancellation between shortwave and longwave heating. The "slanted" pattern of CRH in the extratropics indeed is one reason for sampling CRH in terms of air temperature. No changes made.

Line 170: warming near surface seems not as noticeable from Li et al. (https://doi.org/10.1175/JCLI-D-14-00825.1)?

We are not sure what the reviewers refers to. Li et al. is based on a single model (IPSL-CM5A-LR), while we are looking at 20 models. In the model studied in Li et al., the positive CRH near the surface is not strongly visible (see their Fig. 4). We suspect this is because Li et al. use a non-linear pressure axis that squashes the near-surface atmosphere. No changes made.

Line 177: Is this difference due to cloud difference or other variables? Perhaps good to overlay these plots with cloud amount climatology.

This is difficult to say, and could be due to difference in cloud cover, cloud ice and cloud liquid content, and radiative treatment of clouds (e.g., optical properties of ice crystals). That being said, we cannot trace the sources of the model differences here. Nevertheless, we have added the new Fig. 2 for cloud fraction in the amip simulations and panel f of Fig. 3 of the revised version for observed cloud cover. We have also added the new Subsect. 2.4, which mentions that differences in cloud-radiative heating are not simply explained by differences in cloud fraction (L193-196 of the tracked-changes version).

Line 188: what's causing their difference, given they're based on the same active sensors? Any systematic bias in obs, e.g., due to their limited time/space sampling, when compared to GCMs?

The difference in the two observation-based estimates of CRH from 2B-FLXHR-LIDAR and CCCM is indeed somewhat sobering. The reasons for these differences have been studied in detail by Ham et al. (2017). We have added a new paragraph in Sect. 3 and have included the most recent release R05 of 2B-FLXHR-LIDAR, for which we have calculated cloud-radiative heating. Please see our changes in Subsect. 2.3, where we introduce the satellite-based estimates of cloud-radiative heating as well as L232-242 of the tracked-changes version.

Line 227: beta seems a crucial parameter in this analysis. More explanation and discussion on how this value is set would be helpful.

Agreed. We have reworked this part of the manuscript in Sect. 4 accordingly. Please see our changes in L273-287 of the tracked-changes version, as well as our answer to a similar comment by reviewer 1.

Line 247: "very similar" sounds subjective to me.

In 5 of 7 models, the CRH response is less than 0.1 K/day when sampled as a function of atmospheric temperature and apart from the boundary layer in Fig. 6. Compared to Fig. 4, which samples the response as a function of pressure, this is indeed "very similar" in our view. We acknowledge that this is a subjective statement, but is not any assessment subjective at some level? Still, we have replaced "very" by "remarkably". Please see L306 of the tracked-changes version.

Line 292/298: there seems latitudinal difference which may be a different aspect of Ts control?

We are not sure what the reviewer refers to. No changes made.

**Literature review**

The introduction and comparison of results to previous works would benefit from a more complete inclusion of relevant papers, such as: Zhang et al. (2017), https://doi.org/10.1007/s00382-016-3501-0 and Kato et al. (2019), https://doi.org/10.1029/2018JD028878.

Thanks, we have added these papers, as well as the recent work of Luo et al. (2023). See L49, L58, L73 and L164 of the tracked-changes version.

**References**

Bertrand, L., J. E. Kay, J. Haynes, and G. de Boer, 2024: A global gridded dataset for cloud vertical structure from combined CloudSat and CALIPSO observations. Earth Syst. Sci. Data, 16 (3), 1301–1316, doi:10.5194/essd-16-1301-2024.

Ham, S.-H., et al., 2017: Cloud occurrences and cloud radiative effects (CREs) from CERES-CALIPSO-CloudSat-MODIS (CCCM) and CloudSat radar-lidar (RL) products. J. Geophys. Res. Atmos., 122 (16), 8852–8884, doi:10.1002/2017JD026725.

Luo, H., J. Quaas, and Y. Han, 2023: Examining cloud vertical structure and radiative effects from satellite retrievals and evaluation of CMIP6 scenarios. Atmos. Chem. Phys., 23 (14), 8169–8186, doi:10.5194/acp-23-8169-2023.

Singh, M. S. and P. A. O'Gorman, 2012: Upward Shift of the Atmospheric General Circulation under Global Warming: Theory and Simulations. J. Climate, 25 (23), 8259–8276, doi:10.1175/JCLI-D-11-00699.1.

Webb, M. J., et al., 2017: The Cloud Feedback Model Intercomparison Project (CFMIP) contribution to CMIP6. Geosci. Model Dev., 10 (1), 359–384, doi:10.5194/gmd-10-359-2017.

---

## Author Response (AR2)

**Atmospheric cloud-radiative heating in CMIP6 and observations, and its response to surface warming - Response to Reviewers 2**

**Aiko Voigt, Stefanie North, Blaz Gasparini, and Seung-Hee Ham**

We thank the reviewers for their second round of thoughtful evaluations. We are particulary grateful to reviewer 2 for their valuable comment on "cloud masking." In fact, this was a misunderstanding from our side and we have revised the manuscript to properly address this point, as detailed below.

Reviewer comments are in bold, our response is normal font. Quotes from the revised manuscript are in blue (line numbers refer to the tracked-changes version).

We also found an error in figure reference that we have corrected (L194).

**Reviewer 1**

**I commend the authors on a nice paper and recommend that it be accepted at this time.**

Thank you!

**Reviewer 2**

**The authors addressed most of my comments and I appreciated that. I think the paper has significantly improved as a result.**

Thank you.

**I do still have some questions/comments about the "masking effect". I'd like to note again that CRH, as defined as all-sky and clear-sky heating rate difference, is unable to accurately represent CRH change caused by cloud change, because it potentially aliases the masking effect of cloud "existence" as the effect of cloud "change". Do the cited/added results (Eq. 5 and Fig 4) really rule out this potential aliasing issue? I don't think so. These results showed the variance of all-sky heating isn't explained by clear-sky (which I agree), but can't tell if the dominant cloud term (sigma_cloud) is really due to cloud change or cloud existence (the "masking effect"). One must decompose the CRH term to know if, or to what extent, the masking effect is an issue. Although it is probably beyond the scope to do this decomposition within this paper, this issue should be properly acknowledged by referencing the relevant literature and (e.g., heating rate kernels). This paper can also motivate future work to elucidate and mitigate the issue - with regard to this, I am unclear why the authors, in their reply, seem to suggest it is not feasible to use the heating rate kernels for this purpose. Please clarify.**

Agreed! We have revised the manuscript to discuss the issue of cloud masking and added references to Soden et al. (2004), Huang and Huang (2023) and Huang and Huang (2024). The latter manuscript is currently in review and particularly relevant to our work.

L97-104 (Introduction): Because cloud-radiative heating is defined as the difference between all-sky and clear-sky radiative heating, it depends not only on the cloud field itself but also the clear-sky background state of the atmosphere, an effect known as "cloud-masking" (Soden et al., 2004; Huang and Huang, 2024). Differences in cloud-radiative heating between models, between models and observations, or between different climate states may thus be influenced by non-cloud fields such as temperature and water vapor. These

clear-sky effects could be quantified by explicit radiative transfer calculations or radiative kernel methods, but given the paucity of studies on atmospheric cloud-radiative heating we leave such refinements to future work.

L188-189 (Sect. 2.4): ... and may be affected by cloud masking (cf. Sect. 1).

L223-225 (Sect. 3): Future work that takes into account cloud-masking effects would be helpful to quantify the sources of model differences in all-sky and cloud-radiative heating, for example by means of radiative kernels (Huang and Huang, 2023, 2024).

**References**

Huang, H. and Y. Huang, 2023: Radiative sensitivity quantified by a new set of radiation flux kernels based on the ECMWF Reanalysis v5 (ERA5). Earth Syst. Sci. Data, 15 (7), 3001–3021, doi:10.5194/essd-15-3001-2023.

Huang, H. and Y. Huang, 2024: Diagnosing atmospheric heating rate changes using radiative kernels. doi:10.22541/essoar.171828386.61901229/v1.

Soden, B. J., A. J. Broccoli, and R. S. Hemler, 2004: On the Use of Cloud Forcing to Estimate Cloud Feedback. J. Climate, 17 (19), 3661 – 3665, doi:10.1175/1520-0442(2004)017¡3661:OTUOCF¿2.0.CO;2.